

# Evaluating Post-Wildfire Debris Flow Rainfall Thresholds and Volume Models at the 2020 Grizzly Creek Fire in Glenwood Canyon, Colorado, USA

Francis K. Rengers[1], Samuel Bower[2], Andrew Knapp[3], Jason W. Kean[1], Danielle W. vonLembke[1],
Matthew A. Thomas[1], Jaime Kostelnik[1], Katherine R. Barnhart[1], Matthew Bethel[4], Joseph E. Gartner[5],
Madeline Hille[4], Dennis M. Staley[6], Justin Anderson[7], Elizabeth K. Roberts[8], Stephen B. DeLong[9], Belize
Lane[10], Paxton Ridgway[10], Brendon P. Murphy[11]

[1]Landslide Hazards Program, U.S. Geological Survey, Golden, CO 80401, USA

[2]Department of Geography and Geology, West Virginia University, Morgantown, WV 26506, USA

[3]Colorado Department of Transportation, Denver, CO, 80204, USA

[4] Merrick and Company, Greenwood Village, CO, 80111, USA

[5] BGC Engineering, Inc., Golden, CO, 80401, USA

[6] Landslide Hazards Program,  U.S. Geological Survey, Anchorage, AK, 99508, USA

[7] Tongass National Forest, U.S. Forest Service,  Petersburg, AK, 99833, USA

[8]White River National Forest, U.S. Forest Service, Glenwood Springs, CO, 81601, USA

[9]Earthquake Hazards Program, U.S. Geological Survey, Moffett Field, CA, 94043, USA

[10]Dept. of Civil and Environmental Engineering, Utah State University, Logan, UT, 84322, USA

[11]School of Environmental Science, Simon Fraser University, Burnaby, B.C., V5A 1S6, CA

*Correspondence to*: Francis K Rengers (frengers@usgs.gov)



**Abstract.** As wildfire increases in the western United States, so do postfire debris-flow hazards. The U.S. Geological Survey (USGS) has developed two separate models to estimate (1) rainfall intensity thresholds for postfire debris flow initiation and (2) debris-flow volumes. However, the information necessary to test the accuracy of these models is seldom available. Here, we studied how well these models performed over a two-year period in the 2020 Grizzly Creek Fire burn perimeter in Glenwood Canyon, Colorado, USA, through the development of a debris flow response inventory. The study area had the advantage of a network of 11 rain gauges for rainfall intensity measurements and repeat lidar data for volume estimates. Our observations showed that 89% of observed debris flows in the first year postfire were triggered by rainfall rates higher than the fire-wide rainfall threshold produced by the current USGS operational model (M1). No debris flows were observed in the second year postfire, despite eight rainstorms with intensities higher than the modeled rainfall threshold. We found that the operational model for debris flow initiation rainfall thresholds works well in this region during the first year but may be too conservative in year 2 due to vegetation recovery and sediment exhaustion. However, rainfall thresholds in the second year can be improved by using updated remote sensing imagery to recalculate the debris-flow initiation probability with the M1 model. The current volume model overpredicts for this region by a median value of 4.4 times. However, the offset between the predictions and observations is linear, and the volumes from the Grizzly Creek debris flows had a similar magnitude to historic postfire debris flows in the region. Consequently, the current volume model could be adjusted with a regional correction factor.




## 1 Introduction

Wildfire has been increasing in the western U.S. in recent decades (Westerling et al., 2006; Westerling, 2016), as a warming
climate has increased the number of days of high fire danger (Abatzoglou et al., 2021). This increased fire activity has resulted
in more acreage burned (Dennison et al., 2014) at higher severity (Mueller et al., 2020). Despite this recent spike in activity,
historical records suggest that the western U.S. has a higher potential for wildfire than has been experienced in recent years
(Murphy et al., 2018). Wildfire activity results in a variety of postfire hazards (Santi and Rengers, 2020) from rockfall (De
Graff et al., 2015; Graber and Santi, 2023; Guasti et al., 2013; Melzner et al., 2019) to flash flooding (Brogan et al., 2017;
Brooks et al., 2009; Cannon et al., 2008; Meyer et al., 2001; Neary and Gottfried, 2002; Warrick et al., 2022) to debris flows
(Kean et al., 2019, 2016, 2011; Klock and Helvey, 1976; Murphy et al., 2019; Nyman et al., 2011; Parise and Cannon, 2012;
Santi et al., 2008; Thomas et al., 2021; Tillery and Rengers, 2019; Wells, 1987; Wohl and Pearthree, 1991).

The proximate cause for increased hazards after wildfire in steep, vegetated terrain is the consumption of vegetation by fire
and fire-induced changes to forest hydrology. In pre-fire conditions, the vegetation canopy intercepts incoming rainfall, both
reducing the kinetic energy imparted by rain to the soil and storing some of the water (Rutter et al., 1975, 1971). In addition,
vegetation stems and vegetation litter/duff on the forest floor create hydraulic roughness that can slow and store overland flow
(Arcement and Schneider, 1989). Wildfire changes these characteristics, reducing canopy interception (Williams et al., 2019),
surface roughness (e.g., Hoch et al., 2021; Tang et al., 2019a), and litter/duff water storage (e.g., Ebel, 2013) creating faster
pathways from rainfall to runoff. In addition, soil in many burn areas shows increased water repellency resulting from hyper-
dry conditions (Moody and Ebel, 2012), hydrophobicity (DeBano, 2000; DeBano et al., 1979), or soil pore clogging (Larsen
et al., 2009). These wildfire-induced changes result in more frequent overland flow than in unburned forest conditions. At high
velocities, overland flow can develop sediment transport conditions sufficient to initiate runoff-generated debris flows (e.g.,
Rengers et al., 2019; Tang et al., 2019b).

Rainfall-intensity thresholds can be used to successfully assess the occurrence of postfire runoff-generated debris flow hazards
(Cannon et al., 2008). In particular, short-duration rainfall intensities (<15 min.), which are a key predictor of runoff generation,
have been shown to be the most likely to initiate debris flows (Kean et al., 2011; Thomas et al., 2023). The current USGS
operational model (the M1 model described further in the methods section) predicts the probability of debris-flow initiation
based on short-duration rainfall intensity, burn severity, slope steepness, and soil erodibility (Staley et al., 2017). Using the
M1 model it is possible to estimate a rainfall intensity threshold based on a debris-flow initiation likelihood (e.g., 50%) (Staley
et al. 2017). Spatially explicit rainfall intensity thresholds can be modeled throughout the burn perimeter, but the median
rainfall intensity threshold at a 50% likelihood for all watersheds over the entire burn perimeter is typically used as guidance
to support early warning operations in the first year postfire (U.S. Geological Survey, 2022). The present study leverages an





opportunity to test the performance of the M1 model rainfall threshold and its implementation for operational postfire hazard assessments.

Similar to the model for debris-flow initiation, the USGS operational hazard assessment uses an empirical model to estimate postfire runoff-generated debris-flow volumes. This model incorporates rainfall intensity, burn severity, and topography (Gartner et al., 2014). The volume model was developed using empirical data from southern California, and it has been shown to work well in that region where a large proportion of the debris-flow sediment is sourced from hillslopes (Rengers et al., 2021). In different cases, postfire debris flows in the Rocky Mountains have been observed to incorporate the bulk of their

material from rilling and sheetwash (Cannon et al. 2001) and from channel incision (Santi et al 2008). This raises the question of the applicability of the USGS volume model in the Rocky Mountain region of Colorado, USA.  Accurate estimates of debris-flow volume are particularly important for debris-flow runout models (Barnhart et al., 2021) and the design of debris-flow mitigation structures (Prochaska et al., 2008).

Opportunities to test the current USGS models for debris-flow initiation and volume are rare because they require a relatively

dense rain gauges network (e.g., sufficiently dense to capture small convective rainstorms) and field observations to attribute debris-flow activity and/or volume to individual storms.  The recent Grizzly Creek Fire in August 2020 created a suitable case for model testing in Glenwood Canyon, CO, which is in a region where postfire debris flows have been previously observed (Cannon et al., 2001, 2008). Using the Grizzly Creek Fire, we examined the regional applicability of the two current USGS operational models for debris-flow initiation and debris-flow volume with: (1) a detailed inventory of storms that produced

debris flows versus flood or no response, (2) a dense rain gauge network, (3) pre- and post-event lidar, and (4) airborne and satellite imagery. This work tested the hypothesis that the current USGS operational models for debris-flow rainfall thresholds and volume are applicable at our study site during the first two years following wildfire. We additionally recorded some of the major infrastructure and water resource impacts due to the postfire debris-flow activity.

## 2 Study Area

### 2.1 Wildfire and Site conditions

The Grizzly Creek Fire ignited on 10 August 2020 and burned 13,000 *ha* and 19 river *km* through Glenwood Canyon, CO, until it was fully contained on 18 December 2020. Glenwood Canyon is a narrow, high-relief canyon along the Colorado River. The fire burn severity (Parsons et al., 2010) was a mosaic of high (12%), moderate (43%), and low or unburned severity (45%), with the highest burn severity on the steep canyon slopes near the Colorado River and decreased severity as the fire moved

upslope towards the canyon rim. This wildfire threatened critical infrastructure including U.S. Interstate Highway 70 (I-70), the Union Pacific Railroad, the Shoshone Hydroelectric powerplant, a Colorado Department of Transportation tunnel security facility, and the Glenwood Springs municipal water supply surface water intakes in Grizzly Creek and No Name Creek watersheds.



Glenwood Canyon has a semi-arid climate with an average annual precipitation of 600 mm (National Oceanic and Atmospheric
Administration, 2023). Precipitation is typically composed of snowfall in the winter months and rainfall during summer
thunderstorms, primarily driven by the North American Monsoon, which occurs from June through September (Adams and
Comrie, 1997). The 2021 North American Monsoon was particularly active in western Colorado with precipitation reaching
130-170% of the average annual precipitation (Castellano et al., 2021). Steep cliffs in the canyon create large areas with
minimal soil development (Graber and Santi, 2023, 2022), primarily leading to shallow loams with varying amounts of clay
and gravel (Soil Survey Staff, Natural Resources Conservation Service, United States Department of Agriculture, 2021). The
vegetation at the site includes Pinyon-Juniper Woodlands (*Pinus Edulis* and *Juniperus scopulorum*), Montane Forest and
Shrublands (e.g., *Artemisia tridentata tridentata*, *Sarcobatus vermiculatus*), lodgepole pine (*Pinus contorta*), Douglas-fir
(*Pseudotsuga menziesii*), Engelmann spruce (*Picea engelmannii*), Gambel oak (*Quercus gambelii*), Aspen (*Populus
tremuloides*), and Subalpine Spruce-Fir Forests (United States Forest Service, 2020). The geology of the Glenwood Canyon is
characterized by biotite granites intruding into mica schists and gneisses (Paleoproterozoic age) near the Colorado River
overlain by limestones, marine shales, and gypsum (Kirkham et al., 2009). Finally, there are numerous quaternary-age
landslides within Glenwood Canyon, including a large portion of the Devil's Hole watershed (Kirkham et al., 2009) (Fig. 1).

## 2.2 Debris-Flow Events: Historical and Grizzly Creek Fire

Postfire debris flows have been observed previously near Glenwood Canyon, CO (Cannon et al., 2001, 2008). The South
Canyon Fire in July 1994 burned west of Glenwood Springs, CO and a rainstorm two months later (1 September 1994)
triggered runoff-generated debris flows in the Maroon Formation (Permian-Pennsylvanian aged) (Cannon et al., 2001). The
debris flows trapped 30 cars on I-70, and two vehicles were moved into the Colorado River. The debris flows inundated 0.13
km$^2$ with a volume of 68,000 m$^3$. Field mapping after this event shows that the majority of debris flows were runoff generated
(Cannon et al., 2001). Similarly, the Coal Seam Fire burned 4941 hectares west of Glenwood Springs, CO in June 2002
(Cannon et al., 2008), and debris flows originated within the burn perimeter on 5 August 2002. The peak 10-minute rainfall
intensities at the Coal Seam fire varied between 19.8-57.9 mm h$^{-1}$ during the months following the wildfire (August and
September 2002). Those rates were equal to or less than the 2-year recurrence interval storm, and during these storms a train
and a passenger vehicle were buried during debris-flow events (Cannon et al., 2003, 2008).

Postfire debris flows triggered from June to August 2021 in the Grizzly Creek burn perimeter caused major disruptions in
Glenwood Canyon to roads, railroad lines, the Colorado River (a source of drinking water, energy generation, and a recreation
industry), and a bicycle path adjacent to I-70. Initial repair costs for the highway were more than $50 million (Stroud, 2021a),
and the overall repair costs to road infrastructure were estimated at $116 million (Stroud, 2021b). In addition, debris-flow
disruptions leading to closures of I-70 resulted in lost revenue from interstate commerce, estimated at approximately $1 million
per hour (Erku, 2023). During the summers of 2021 and 2022, there were a combined total of 14 road closures (Colorado
Department of Transportation [CDOT], personal communication). Railroad closures of passenger and freight rail due to debris
flows caused severe disruptions (Erku, 2021a), and the commercial rafting and recreation industry in Glenwood Canyon was



substantially affected (Blevins, 2021). Finally, long-term degradation of water quality was caused by debris-flow material deposited in the Colorado River. Communities more than 30 km downstream of Glenwood Canyon required enhanced filtration for turbidity and heavy minerals for months following the debris flows (Erku, 2021b).


## 3 Methods

### 3.1 Hazard Assessment

The U.S. Geological Survey produces hazard assessment maps to provide early warning in burn areas in the United States (e.g., U.S. Geological Survey, 2022). These hazard assessments contain information on debris-flow initiation likelihood (a metric that can be used to estimate a rainfall triggering threshold), debris flow volume, and a combined hazard (Staley et al., 2017). The M1 model used to estimate the likelihood of debris flow initiation ($p$) (Figure 1) contains four primary inputs: burn severity (Parsons et al., 2010) (a measure of soil/vegetation change), slope, change in normalized burn ratio (dNBR), and the soil KF-factor (a measure of erodibility) obtained from (Schwartz and Alexander, 1995). this can be used to determine a rainfall rate expected to trigger debris flows using:


$$R_{(p)} = \frac{ln\left(\frac{p}{1-p}\right) - \beta}{C_1T + C_2F + C_3S} \qquad \textit{Equation 1}$$

where $R(p)$ is the peak rainfall accumulation [mm] over a time duration (here we use a 15 minute duration), $\beta$ is the intercept of the M1 logistic regression model, T is the proportion of upslope area with moderate to high burn severity and gradients $\geq$ 23°, F is the average dNBR of upslope pixels divided by 1000, S is the average KF-Factor of upslope area, C1, C2, and C3 are empirically defined coefficients.

The Year 1 and Year 2 rainfall thresholds in the USGS operational hazard assessments are estimated from Equation 1 assuming p = 50% (P50) and p = 75% (P75) likelihood, respectively (Staley et al., 2017). These probabilities are used to define a 15-minute intensity rainfall threshold ($\overline{I15_T}$); however, the success rate of the P50 and P75 thresholds have not been rigorously tested. Therefore, in this study we compared the median (P50) $\overline{I15_T}$ for all basins in the burn perimeter with the measured peak 15-minute intensity (I15) in 2021 ($\overline{I15_{T21}}$) and the P75 in 2022 ($\overline{I15_{T22}}$) to determine the performance of the P50 and P75 thresholds estimated using Equation 1. We used a single threshold for the entire fire, as opposed to using separate I15 thresholds for individual channel segments or basins because, in practice, managers generally use a single threshold value as the basis for issuing warnings.

The volume model used in USGS operational postfire debris flow hazard assessments was developed by Gartner et al. (2014), with the form:

$$ln(V_p) = 4.22 + 0.39\sqrt{I15} + 0.36ln(Bmh) + 0.13\sqrt{R} \qquad \textit{Equation 2}$$




where $V_p$ is volume [m³], I15 is the 15-min rainfall intensity [mm h⁻¹], Bmh is watershed area burned at moderate and high severity [km²], and R is the watershed relief [m]. This approach was developed using postfire debris flow volume data from sediment retention basins in southern California (Gartner et al., 2014).

### 3.2 Rainfall Monitoring Network

Three tipping bucket rain gauges were installed in the burn area on 17 September 2020 by the USGS Landslide Hazards Program. These gauges were deployed specifically for the task of verifying the hazard assessment model (Figure 1). An additional seven rain gauges (7) were added during the summer of 2021 by the USGS Colorado Water Science Center to provide situational awareness and inform operational warnings by the National Weather Service and CDOT. We were also

able to obtain data from the Bair Ranch gauge operated by CDOT. As a result, 11 total rain gauges captured high frequency (15-minute) rainfall intensities in and around Glenwood Canyon (Figure 2), and provided a relatively dense network of rain data to associate with debris flow events. All gauges were operating by mid-July 2021, beginning on several different dates (Table 1).

### 3.3 Inventory of Debris Flows and Storms

We generated an inventory of storms and debris flows using the rain gauge network within the fire perimeter (Rengers et al., 2023b). Observations of debris flows in drainages along I-70 were provided by CDOT personnel following storms. We used rainfall data to identify specific storms over the canyon during the 2021 and 2022 monsoon seasons, defining each new storm as the first measurement of precipitable water following a period of more than 8 hours without rainfall (Staley et al., 2013). This standard was used to maintain consistency with Staley et al., (2017). For each defined storm, we calculated the total

rainfall, the storm duration, and the peak 15-, 30-, and 60-minute rainfall intensities (Rengers et al., 2023b).

We associated each debris-flow occurrence with rainfall data from a nearby representative rain gauge. Several rules were used to choose which rain gauge to attribute to a given debris-flow observation. First, we were limited by the available rainfall record, so we only used gauges with a rainfall record spanning the storm event (Table 1.). Second, we attempted to account for channel network connectivity. That is, the Colorado River dissects Glenwood Canyon in a general east-west direction, and

tributary drainages flow obliquely into Colorado River from the North and South rims of the canyon (Figure 2a). Consequently, if a debris-flow observation was at the mouth of a drainage on the north side of the Colorado River, we would prioritize using a rain gauge on the north side of the canyon because the rainfall was expected to be more reflective of the upstream drainage basin that contributed to the observation rather than selecting a closer gauge on the opposite side of the Colorado River. Finally, if there were multiple rain gauges within a similar distance to the observation, we used a conservative approach and associated

the gauge with the highest I15 rainfall record to the debris-flow observation. The maximum distance between an observation and its associated gauge was 6.8 km, the minimum distance was 0.2 km, and the average distance was 4.2 km (Rengers et al.,



2023b).  These distances are similar to the rules used by Staley et al., 2016 (rain gauges within 4 km$^2$) in generating the debris flow inventory used to develop the M1 model.

## 3.4 Debris Flow Initiation and Volume

### 3.4.1 Mapping Debris Flow Initiation

Low-altitude aerial imagery and lidar were collected from a crewed aircraft on 24 August 2021, and we used these data to map points of debris-flow initiation. Unlike many runoff-generated debris flows that initiate from coalescing rills (e.g., Tillery and Rengers, 2019), debris-flow initiation at this site occurred primarily within channels (Figure 4). We mapped initiation points where a difference in channel scour was visible in the imagery. Select mapped locations were verified with field observations in the Blue Gulch (Fig. 3), French Creek, and Grizzly Creek watersheds. Mapped debris-flow initiation points were subsequently checked against a before-after lidar Digital Elevation Model (DEM) of difference (DoD) (see section 0). If the DoD showed a distinct change from no erosion to erosion at a mapped debris-flow initiation location, the mapped initiation point was retained, but if there was no change in erosion in the DoD, the initiation point was rejected.

### 3.4.2 Lidar Collection and Volume Estimation

Lidar data were available before and after the Grizzly Creek Fire in 2016 and in 2021 (Rengers et al., 2023a). The 2016 lidar was collected during a series of flights between 10 June and 7 October 2016. The 2021 lidar flight was conducted in full on 24 August 2021. An initial investigation of the 2016 point cloud determined that there was an internal flightline offset of approximately 0.5 m.  To address this, we requested that the original vendor re-process the 2016 point clouds. In order to determine any misalignment between the 2016 and 2021 datasets, we examined the distribution of elevation differences over a large low-slope hillslope area of the canyon and found that the change was normally distributed around a mean of 0 (Fig. S1). This analysis shows that there was no systematic horizontal offset in the two datasets. We evaluated the level of detection (LoD) for the lidar difference along I-70, assuming that the road surfaces were stable and fixed between the 2016 and 2021 scans. The paved roads consistently showed error variance less than 10 cm, therefore we used +/- 10 cm as our LoD (Fig. S2). Using the two lidar datasets (1 m pixel), we created a DoD map of erosion and deposition throughout the Grizzly Creek burn area (Figure 2 and Figure 3). For each debris-flow observation in our inventory, we mapped erosional and depositional areas in each channel with separate polygons. We masked cliff areas from the DoD and all slopes > 45° to eliminate lidar artifacts due to interpolation across steep slopes.  Additionally, some spurious change was observed on the hillslopes, likely due to vegetation classification errors, however this was eliminated by focusing our analysis to channelized areas where field observations showed the primary debris flow activity to occur.   The mapped channel polygons allowed us to quantify the provenance of the debris-flow material and its volume as a function of contributing drainage area, as well as to examine the erosion to deposition transition in debris-flow channels.





The eroded volume was estimated by summing DoD pixels within each mapped erosion polygon from the estimated point of debris-flow initiation to the onset of deposition. Similarly, deposited volume was estimated by summing DoD pixels within the deposition polygon from the transition point to the base of the fan. At sites where the depositional fan was retained, we
compared the volume of material eroded upstream to the volume of material deposited. When fans were not modified by fluvial erosion or highway cleanup work, the upstream erosional measurements in the DoD should match the downstream depositional measurements within the LoD uncertainty. Confidence in the erosion estimates from the DoD is relatively high because the channels were mostly undisturbed between the debris-flow activity and the post-event lidar flight. However, there was more uncertainty around depositional areas.


### 3.4.3 Fan Volume Estimate

Debris flows triggered during the 2021 storms deposited sediment as either (1) large fans at watershed outlets, or (2) in-channel fans. The timing of the large fans at channel outlets was captured in our inventory because they were observed by CDOT personnel during the event in the canyon. The timing of in-channel fan deposition was more complicated because there were
not witnesses to document the timing. In-channel fans were specifically observed in French Creek, Tie Gulch, Grizzly Creek, and Deadhorse Creek (Figure 5). Debris flows observed along Grizzly Creek most likely occurred along 31 July 2021 based on review of 3 m resolution satellite imagery (Planet Labs, 2018). A storm on this date also caused debris flows elsewhere in the burn area. The remaining three drainages (French Creek, Tie Gulch, and Deadhorse Creek) with in-channel deposits could not be assigned to a single storm, due to gaps in imagery and cloud cover. The deposit in French Creek likely occurred between
3-5 July 2021 based on trail camera data (Video S1) and satellite imagery (Planet Labs, 2018). The deposit in Deadhorse Creek likely occurred between 5-12 August 2021 based on satellite imagery (Planet Labs, 2018). The timing of debris-flow deposition in Tie Gulch could not be identified specifically, but it occurred in the summer of 2021.

Large fans at watershed outlets were identified in six locations along the Colorado River (Table 1). Of the six fans where timing was known, five were triggered by a storm on 31 July 2021. A sixth fan at Devil's Hole resulted from two separate
storms that caused debris-flow deposition: a storm on 22 July 2021 and a second storm on 29 July 2021 (Figure 2c). Note that all debris-flow fans deposited onto I-70 were removed by maintenance crews prior to the 2021 lidar collection, and therefore fan deposition on I-70 was not available for analysis.

To estimate the depositional fan volume of the six fans in the Colorado River, we divided the deposit into three sets of volumes ($V_1$, $V_2$, and $V_3$) (Figure 6). The volume in $V_1$ is the total volume of sediment deposited upstream of the Colorado River
computed from the DoD. This volume is calculated as:

$$V_1 = \sum_{i=1}^{n} (Z_{2021} - Z_{2016}) A_{cell} \qquad\qquad Equation\ 3$$



where i represents the index of each DoD cell in a deposited fan, n is the total number of cells in the DoD of the deposited fan, $Z_{2021}$ is the elevation of the lidar DEM from 2021, $Z_{2016}$ is the elevation of the lidar DEM from 2016, and $A_{cell}$ is the surface area of the DEM cell. Next, we divided the volume deposited in the Colorado River into two pieces. $V_2$ is the subaerial volume above the river water surface elevation (WSE) at the time of the post-event lidar collection. $V_2$ was calculated using:

$$V_2 = \sum_{i=1}^{n}(Z_{2021} - h_w)A_{cell} \hspace{3cm} Equation\ 4$$

where $h_w$ is the average elevation at the margins of the fan, which serves as a proxy for the WSE at the time of the 2021 lidar flight. Finally, $V_3$ represents the volume underneath $h_w$. Here we used personally communicated reports of the average sediment depth (d) for the entire plan view mapped area below the water surface ($A_{planview}$) estimated from CDOT maintenance crews who excavated the material (Table 2).

$$V_3 = A_{planview}d \hspace{3cm} Equation\ 5$$

Sediment deposited in the Colorado River was removed both by fluvial erosion and by mechanical excavation intended to protect infrastructure. Because the post-event lidar was collected after some of this removal already occurred and because, pre-event bathymetry of the Colorado River was unknown, we expect our approach gives lower bound estimates for depositional fan volumes within the Colorado River.

### 3.4.4 Volume Model Analysis

We compared the observed erosional debris-flow volumes with the current debris-flow volume model used in USGS postfire debris-flow hazard assessments (*Equation 2*). For this comparison, we selected the transition point of erosion to deposition in drainage basins where debris-flow timing was known to calculate Bmh and R. Debris-flow timing recorded in our inventory was used to select the peak I15 from the nearest representative rain gauge. If there were multiple storms that were triggered in the same watershed, we used *Equation 2* to estimate $V_p$ from each storm, and then summed volumes. We fit a regression line to the $V_p$ versus $V_o$ to examine the success of the predicted volumes and examined how $V_p$ and $V_o$ changed as a function of upstream drainage area.

To set the Grizzly Creek Fire debris flows volume observations into regional context, we also compared the Grizzly Creek volume data to available volume data from debris flows observed following the South Canyon and Coal Seam fires. Limited rainfall data for the South Canyon Fire precluded the use of *Equation 2*, however upstream drainage area and debris-flow volume data were available (Cannon et al., 2001). The Coal Seam Fire had sufficient data to perform a comparison between observed volumes and predicted volumes computed using *Equation 2*.





### 3.5 Vegetation Recovery

Vegetation recovery was monitored by field excursions in the burn area to qualitatively observe regrowth within specific plant communities, in particular Gambel oak (*Quercus* gambelii), cottonwood (*populus angustifolia)*, Aspen (*Populus tremuloides*), and mixed conifer that include Douglas-fir (*Pseudotsuga menziesii*).  To quantify postfire regrowth of vegetation and evaluate its effect on debris-flow susceptibility in the Grizzly Creek burn area, we examined three remotely sensed satellite vegetation indices. We acquired satellite imagery from Landsat 8-9 (Collection 2 – L2 processing level, 30 m spatial resolution, 8 to 16-day revisit time) and Sentinel 2 (Level 2A processing level, 10 m spatial resolution, 5-day revisit time).  The imagery was used to quantify changes in surface reflectance-derived spectral indices, which represent vegetation states across the Grizzly Creek Fire (Sentinel Hub, 2022).  With these data we calculated the Normalized Burn Ratio (NBR; near infrared and short-wave infrared wavelengths), Normalized Difference Vegetation Index (NDVI; red and near infrared wavelengths), and the Enhanced Vegetation Index (EVI; red, blue, and near infrared wavelengths). Vegetation index values were averaged across each modeled basin (U.S. Geological Survey, 2022) that intersects the burn perimeter. We then tracked these indices during five distinct periods: (1) at the beginning of the monsoon season (15 June 2020) prior to the fire, (2) prior to fire ignition (10 August 2020), (3) after fire containment (18 December 2020), (4) at the beginning of the monsoon season in 2021 (15 June 2021), and (5) at the beginning of the monsoon season in 2022 (15 June 2022). Owing to differences in satellite revisit times and atmospheric conditions over the burn area, the Landsat and Sentinel imagery collection times are typically within two weeks of each other. We estimated recovery as the increase in reflectance in vegetation indices from the postfire period divided by the difference in reflectance between the pre and postfire periods,

### 3.6 Using dNBR to Estimate a Year 2 Threshold

We tested the current method of using P75 as a rainfall threshold for year 2 against a new approach using remotely sensed metrics.  As an alternative to P75, we re-calculated the hazard assessment (Equation 1) using the dNBR from the Landsat imagery at the beginning of the second monsoon season and estimated rainfall thresholds based on the recovered vegetation indices. We compared the median (P50) approach using the updated dNBR values in Equation 1 and the P75 estimated from the original dNBR with the observed rainfall rates and debris-flow activity in year 2.  This allowed us to test the efficacy of the current P75 approach.



## 4 Results

### 4.1 Predicted Rainfall Thresholds

The debris-flow inventory reported by CDOT staff in the summer of 2021 captured 40 debris flows in 25 drainages (Rengers et al., 2023b). Some of these locations produced debris flows during more than one storm (Figure 2 and Figure 7). There were no debris flows reported in 2022. The storm inventory during the summer of 2021 included 49 rainstorms with a peak I15 greater than 1 mm h$^{-1}$ from 15 June 2021 to 29 September 2021. Nine of these storms triggered one or more debris-flows (Figure 8a). The storm inventory during the summer of 2022 included 56 unique rainstorms with a peak I15 greater than 1 mm

h$^{-1}$ from 18 June 2022 to 17 September 2022 (Figure 8b).

When the debris flow and storm inventories were compared with the P50 rainfall threshold created by the debris-flow hazard assessment in 2021, we observed relatively good performance of the estimated rainfall threshold (Figure 8a). The I15 values associated with initiation of one or more debris-flows (Figure 8a), were above the $\overline{I15_{T21}}$ (25.9 mm h$^{-1}$) produced by the hazard assessment (Staley et al., 2017) in 8 out of 9 cases during the first year postfire (U.S. Geological Survey, 2022). The only

observed debris flow with a rainfall rate below $\overline{I15_{T21}}$ was on 27 June 2021. However, most of the rain gauges were not operating at that time (Table 1 and Fig. S3); therefore, it is possible that the closest available rain gauge was not reflective of the maximum rainfall rate in the storm.  In the second summer postfire, the $\overline{I15_{T22}}$ was increased (33.7 mm h$^{-1}$) and several storms generated I15 rainfall intensities larger than $\overline{I15_{T22}}$, however, no debris flows were documented (Fig. 7b).

Spatial variability in debris flows occurrence indicated a high degree of spatial variability in the monsoonal storms. Some

storms only produced debris flows within a small portion of the burned area (e.g., 14 July 2021, 22 July 2021), while other storms produced debris flows across a wider area within the burn perimeter (e.g., 29 July 2021) (Figure 7b). No debris flows were observed after 3 August 2021 despite rainfall intensities that exceeded the rainfall threshold. This observation likely results from a combination of vegetation recovery and exhaustion of sediment supply in susceptible basins.

### 4.2. Debris Flow Initiation and Volume


#### 4.2.1 Initiation Mechanisms revealed through Mapping

Field observations and the lidar DoD revealed that the majority of sediment incorporated into debris flows in 2021 originated in channelized areas with relatively minor sediment contribution from adjacent hillslopes (Figure 2b-c and Figure 4). Field observations suggest that channel erosion was nucleated in part by surface water that was strong enough to uproot grassy

vegetation to access sediment below the root zone (Figure 4), as has been observed in grassland settings (Rengers et al., 2016). Moreover, field mapping in multiple watersheds (Blue Gulch, Grizzly Creek, and French Creek) confirm the observations of in-channel debris-flow initiation observed in the DoD. Hillslope rilling was only observed in a few locations with relatively low levels of revegetation in 2021 (Fig. S5).



### 4.2.2 Debris-Flow Volume Observations and Predictions

We mapped 26 debris flows channels along the I-70 corridor and 8 debris flows contained in the Grizzly Creek watershed, and measured a net erosional volume of 460,000±16,700 m³ using the lidar DoD. Because the timing of the flows along Grizzly Creek is less certain (i.e., no personnel were immediately on the scene for clean-up), we do not include the Grizzly Creek flows in the inventory with an associated storm (Rengers et al., 2023b), though imagery suggests they occurred on July 31 2021. The observed debris-flow erosional volumes were substantially smaller than the volume predicted by *Equation 2* (Figure 7 and Figure 9). For example, the observed volume of erosion for individual watersheds ranged from $160 \pm 25$ m³-107,000 m³ $\pm$ 3800 m³, whereas the predicted volumes ranged from 270-470,000 m³. A best-fit line derived from a linear regression was developed to compare the predicted to observed erosion (Grizzly Creek Fire data only):

$$V_o = 0.21V_p + 180 \hspace{4cm} \textit{Equation 6}$$

The depositional volumes observed were less than the erosional volumes, as was expected due to changes between the debris flows and the lidar flight (Figure 7). In a few cases the depositional volume estimates were slightly larger than the erosional estimates, reflecting the large uncertainty in depositional volumes.

The volumes from the Grizzly Creek Fire were similar in magnitude to the observed volumes from the two prior postfire debris flows in near Glenwood Canyon, the Coal Seam and South Canyon Fires (Figure 9), suggesting regional similarities between the observed volumes across three different fires. Moreover, the trend of volume overprediction by *Equation 2* was observed in both the Grizzly Creek Fire data and the Coal Seam Fire data (Figure 9). This suggests that the overprediction of *Equation 2* is not related to fire or storm-specific characteristics of the Grizzly Creek Fire. Rather the overprediction of *Equation 2* is more likely related to regional differences in sediment, vegetation, and sediment transport processes tied to the regional geomorphology.

### 4.3. Vegetation Recovery and Year 2 Threshold

All three satellite vegetation indices show similar declines from the prefire period to the postfire period. After this initial decrease, gradually higher vegetation index values are observed in later epochs (Figure 10). Our median Landsat- versus Sentinel-derived NBR, NDVI, and EVI calculations are well correlated at the basin scale for all calculation periods ($R^2 \geq 0.99$). Recovery levels are consistent across satellite platforms but exhibit some variability across vegetation indices. We calculated 16, 21, and 34 % recovery among NBR, NDVI, and EVI for the first postfire monsoon season and 53, 50, and 68 % recovery among NBR, NDVI, and EVI for the second postfire monsoon season. When averaged across satellite platforms and vegetation indices, we estimate 24 % and 57 % recovery for the first and second postfire monsoon season following the fire, respectively. A new P50 estimate for the second year ($\overline{I15_{T22}}$) using updated dNBR values from Landsat imagery (Landsat Modified Threshold) resulted in an updated rainfall threshold (40 mm h⁻¹). This value was larger than the original P75 rainfall threshold



(33.7 mm h$^{-1}$) (USGS Threshold) eliminating four storms (Figure 8b). Consequently, the new Landsat modified threshold based on measured recovery rates may help to avoid false alarms in the second year postfire.

### 4.4. Infrastructure and Water Resource Impacts

During the collection of our debris flow and storm inventory in Glenwood Canyon, we additionally observed many major
impacts from the debris-flow activity. Debris flows within Glenwood Canyon damaged railroad lines (Union Pacific), roads (I-70), and the Colorado River (Fig. 11). One relatively large debris flow sourced from the Devil's Hole watershed (Fig. 2 and Fig. 3) fully blocked the flow of the Colorado River temporarily on 22 July 2021 (Fig. 11d). The storm on 22 July 2021 started at 4:34 pm (local time) and reached a peak intensity of 21.3 mm h$^{-1}$. The upstream flow gauge on the Colorado River showed a slight increase in flow during the storm, however, the downstream flow gauge at Glenwood Springs shows a drop in river
level (Fig. 11d-e). During the second monsoon season in 2022, despite the lack of new debris-flow observations, we observed erosion of debris-flow fan sediment from 2021 via fluvial erosion (Fig. 12a-b).

### 5 Discussion

The USGS M1 likelihood model successfully estimated an appropriate rainfall threshold for most debris flows in year 1 using the P50 value. For eight out of nine debris-flow-producing storms in 2021, the observed peak I15 from the rain gauges was
greater than the median fire-wide rainfall threshold produced by the hazard assessment model (U.S. Geological Survey, 2022) (Figure 8). This result is similar to previous research (Kean et al., 2011), showing debris flows initiating during storms with high short-duration rainfall intensities. However, the spatial footprint of any given storm observed at our study area was highly variable. For example, during the storm on 22 July 2021, rainfall intensities only exceeded the rainfall threshold at 6 of the 11 rain gauges (Fig. S3), and the observed debris flows from that storm intersected a narrow section of I-70 (between mile marker
124-126) (Figure 7b). Similarly, during the storm on 29 July 2021, rainfall intensities exceeded the rainfall threshold at 8 of the 9 gauges (two gauges had data gaps at this time, Fig. S3), and debris flows were observed throughout most of the canyon (Figure 7b).

Field observations continued through the second year postfire to determine the applicability of the year 2 rainfall threshold ($\overline{I15_{T22}}$). Despite a relatively active monsoon season in 2022 producing high rainfall rates during some storms, no debris flows
were observed in 2022 (Figure 8b). Field observations showed evidence of fluvial reworking of 2021 debris-flow deposits (Figure 12a-b). The 2022 rainfall data suggest that the P75 rainfall intensity threshold predicted by the USGS hazard assessment was exceeded during 8 storms (Figure 8b). However, because no debris flows were observed, it appears that the rainfall threshold associated with the 75% likelihood may have been too conservative for the second monsoon season. We conclude that the rate of vegetation recovery by the second year (57%) along with sediment exhaustion from debris flows in
2021, had greatly reduced runoff and sediment yield by the 2022 monsoon. By contrast, only four of the 2022 storms exceeded



the Landsat modified threshold based on the revised dNBR for 2022 (Figure 8b), but this still does not eliminate all of the false positives. Consequently, more research may be needed to accurately estimate $\overline{I15}_T$ for year 2.

Qualitative field observations of vegetation recovery were aligned with remote sensing observations. After the fire the conifer-dominated stands had some of the worst soil burn severity and least vegetative recovery. Aspen stands and conifer stands with

an aspen component saw vigorous recovery. The Gambel oak, which occupies many of the lower, hotter slopes and had its own vigorous regrowth immediately after the fire, and likely played a role in stabilizing slopes for year 2. Finally, the cottonwood trees were stable points that aided in nucleating debris deposition in Cinnamon Creek, French Creek and Grizzly Creek.

The operational USGS volume model generally overpredicted debris-flow volumes in the Grizzly Creek burn area by a median

value of 4.4 times. The deviation between the model and the observations could likely result from differences between the calibration dataset used to develop Equation 2 and the present study area. Equation 2 was calibrated using data from the Transverse Ranges of southern California, which contains oversteepened hillslopes due to ongoing tectonic activity (DiBiase et al., 2012), that contribute large amounts of sediment to channels through dry ravel (DiBiase et al., 2017; DiBiase and Lamb, 2013; Lamb et al., 2011). By comparison, the Glenwood Canyon formed during the White River Uplift, part of the Laramide

Orogeny, where tectonism ceased 55-35 ma (Allen and Shaw, 2008). The volume of sediment eroded by Grizzly Creek Fire debris flows increased as a function of upstream drainage area (Figure 7), similar to the original observations used to develop Equation 2 (Gartner et al., 2014). Finally, the general trend of larger observed erosional volumes compared to depositional volumes, suggests that future estimates of the erosional volumes can provide a conservative approach for estimating the resulting depositional volume.

Postfire debris-flow volumes from the South Canyon and Coal Seam Fires are of similar magnitude to observed volumes from Grizzly Creek. The volumes from the two historic fires might be expected to be larger than those from Grizzly Creek, because the South Canyon and Coal Seam debris flows were triggered within a few months of the wildfire, and the sediment was derived from both hillslope and channel sources (Cannon et al., 2001, 2008). Moreover, the South Canyon Fire sourced sediment from the Maroon formation, which produces landslides and debris flows even without the influence of wildfire

(Mejía-Navarro et al., 1994). However, the Grizzly Creek debris-flow volumes showed a similar magnitude to the historic volumes even when normalizing by upstream contributing drainage area (Figure 9b), suggesting that there are regional controls on postfire debris-flow volume. Consequently, because of the linear nature of the offset in the volume prediction (Figure 9), it may be possible to apply a linear correction to the predicted volumes (e.g., divide the predicted volume by 4.4) to obtain a regionally corrected volume prediction.

A few large drainages with contributing areas > 9 km$^2$, French Creek, Deadhorse Creek, Tie Gulch, and Grizzly Creek, stored sediment internally without depositing large fans in the Colorado River (Figure 5). In the case of French Creek, a debris flow fan with signs of incision at a drainage area of 15.3 km$^2$ existed prior to the 2021 debris-flows. During debris-flow activity in 2021, this fan aggraded (Video S1). The location of the debris flow deposition may have been influenced by a large concrete retaining wall constructed for a bike path bridge, without which, sediment may have moved into the Colorado River (Figure



5). By contrast, the deposits in Grizzly Creek and Tie Gulch appear to be controlled by natural sediment depositional dynamics. A wide, low gradient valley reach allowed for in-channel fan development in Grizzly Creek at a drainage area of 9.8 km$^2$. Similarly, a low sloping channel section fostered deposition in Tie Gulch at a drainage area of 9.6 km$^2$ (Figure 5). Deposition was also evident at the outlet of Tie Gulch where it debouched onto I-70, but deposition did not reach the Colorado River. Deadhorse Creek showed a mix of minor erosion and deposition throughout the drainage, and minor fan development upstream

of the confluence with the Colorado River at a drainage area of 26 km$^2$. The differences in Deadhorse Creek may be related to the karst geology in the watershed, and Hanging Lake on the East Fork Deadhorse Creek tributary should have reduced the energy of any flows. The internal deposition shown in these larger drainages suggest there may be a drainage area threshold around approximately 10-20 km$^2$, where the morphology of valley floors promotes internal deposition and a transition from debris-flow to debris-flood or hyperconcentrated-flow conditions. This is consistent with observations from prior datasets

showing true debris-flow initiation only in drainage areas < 8 km$^2$ (Staley et al., 2016).

In prior studies, some post-wildfire debris flows source sediment primarily from hillslope erosion, some by channel erosion, and some are balanced by both (Alessio et al., 2021; Nyman et al., 2020; Pelletier and Orem, 2014; Rengers et al., 2021; Tang et al., 2019a). DoDs and field observations suggest that the source of sediment for debris flows in Glenwood Canyon (Figure 2) appeared to be primarily derived from channels. Unlike sites where sediment is sourced primarily from hillslope erosion

(e.g., Rengers et al., 2021), debris flows in the Glenwood Canyon study area initiated primarily in channels at small drainage areas and steep slopes (Figure 4). These differences are likely related to the timing of the debris flows with respect to the wildfire. That is, the fire happened in the fall of 2020, allowing time for partial recovery of soil hydrology during the winter freeze thaw cycles and vegetation regrowth during the spring and early summer of 2021 (Figure 4). While hillslope rilling was observed in select areas (Fig. S5), most hillslopes were partially re-vegetated by the 2021 monsoon based on remote sensing

data (Fig. 9). As a result, flow velocities sufficient to initiate debris flows would likely have only been reached in channels, due to increased hydrologic roughness on hillslopes.

The overall impact of the debris flows on infrastructure following the Grizzly Creek Fire was substantial. There was no loss of life due to any of the debris flows, although there were near-miss instances where people escaped cars surrounded by sediment (Otarola, 2021). At least 30 people in cars were forced to shelter overnight in a tunnel (Stroud, 2021c). Infrastructure

damage included buried railroad lines (Union Pacific), road and bridge damage on I-70, and damage/flooding on the bike path that parallels the highway (Figure 11b-d). Substantial sedimentation in the Colorado River impacted whitewater rafting tourism (Stroud, 2022), and increased overbank flood potential by filling in the riverbed (Figure 12a). The debris flow response following the Grizzly Creek Fire illustrates how debris flows can affect multiple locations during many different storm events, putting different types of critical infrastructure at risk.




## 5 Conclusions

An inventory of postfire debris flows from 2021 and 2022 following the 2020 Grizzly Creek Fire was used to test operational USGS models and methods used for estimating rainfall thresholds and debris-flow volumes. We found that during the first year following the wildfire, the rainfall threshold was successful for the wildfire perimeter as a whole. During the second year following wildfire, no debris flows were observed. The second-year rainfall threshold was exceeded by eight storms using the current operational approach, but when remote sensing data of recovered vegetation was used in the M1 model to generate a second-year threshold only four storms exceeded the threshold. The volume model tended to overestimate the observed volumes by a factor of ~ 4, and a comparison with historic postfire debris flows in the region suggests that reducing predicted volumes by this factor could aid in more realistic volume predictions across the region. More generally, the geomorphic legacy of the 2021 debris flows will likely continue in Glenwood Canyon, as channels scoured to bedrock refill, and debris-flow deposits continue to be reworked by fluvial processes.

## 6 Author Contribution

FR prepared the manuscript, fieldwork, and performed analyses; SB mapped volumes; AK helped develop the debris flow inventory; JWK and DS performed field work; DV MT and JK performed satellite imagery analysis of vegetation recovery; KB performed and developed analysis methods for depositional volume; MB, SD, JG and MH helped to obtain, process, and plan the analysis of the lidar; JA and ER performed field investigations of vegetation and debris flow initiation; BL, PR, and BM investigated field changes fluvial channels from year 1 to year 2 and using remote sensing.

## 7 Competing Interests

The authors declare that they have no conflict of interest.

## 8 Disclaimer

Any use of trade, product, or firm names is for descriptive purposes only and does not imply endorsement by the U.S. Government.

## 9 Code/Data Availability

The data used in this study has been released in two separate data repositories:

Rengers, F.K., Bethel, M., Group, R., Vessely, M., Anderson, S., 2023a. Airborne Lidar Data (2016 and 2021) Airborne Lidar Data (2016 and 2021) Capturing Debris Flow Erosion and Deposition after the Grizzly Creek Fire in Glenwood Canyon Colorado: U.S. Geological Survey data release, https://doi.org/10.5066/P99OT77K.



Rengers, F.K., Bower, S.J., Knapp, A., Kean, J.W., Staley, D.M., Banta, J.R., Williams, C., 2023b. Debris Flow, Precipitation, and Volume Measurements in the Grizzly Creek Burn Perimeter June 2021-September 2022, Glenwood Canyon, Colorado: U.S. Geological Survey data release, https://doi.org/10.5066/P9Z7RROL.


## 10 Acknowledgements

Any use of trade, firm, or product names is for descriptive purposes only and does not imply endorsement by the U.S. Government. We are grateful for internal USGS reviews from Andrew Graber and Jacob Woodard.



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



**Figure 1. USGS Debris Flow Hazard likelihood estimate produced for the Grizzly Creek Fire perimeter using Equation 1. The**
**likelihood of debris flow initiation is shown for a modeled rainstorm with a 15-minute rainfall intensity of 24 mm h⁻¹.**





**Figure 2. (a) Map showing the burn perimeter, rain gauges, and locations of debris flow observations. (b) DEM of difference map showing erosion (red) and deposition (blue) in the lower half of the Blue Gulch drainage. (c) DEM of Difference (DoD) showing erosion (red) and deposition (blue) in the lower half of the Devil's Hole drainage.**




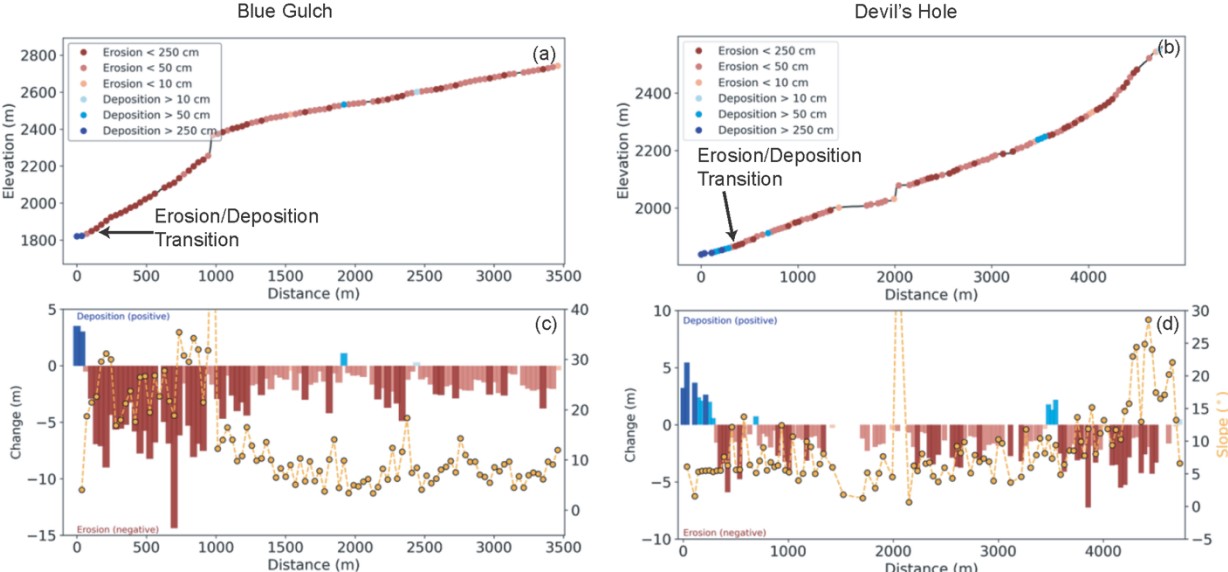

**Figure 3. (a and b) Elevation as a function of distance from the watershed outlet for the Blue Gulch watershed (a) and Devil's Hole watershed (b). Red dots and blue dots represent erosion/deposition, respectively. Areas without a dot did not experience change beyond the level of detection. (c and d) Measured deposition and erosion as a function of distance from the watershed outlet (primary y-axis). Local slope is shown on the secondary y-axis.**






**Figure 4. Debris flows were triggered in the Blue Gulch Watershed on 29 July 2021 and 31 July 2021. Photos from 17 August 2021 show debris flow initiation transitioning from (a) water flow matting down grass (see people for scale), (b) to incipient erosion of grassy rootwads without distinct channel incision, (c) to channelized erosion below the root layer. (d) © Google Earth imagery of the upper portion of the Blue Gulch watershed. (e) Photo from 18 August 2021 of a cliff (101 m relief) in the Blue Gulch watershed 2.2 km downstream of the initiation location. (f) View from the cliff edge of the debris flow path towards the Colorado River.**






**Figure 5. In-channel sediment deposits in large drainage basins. (a) French Creek: a fan forms upstream of the Colorado River, primarily because sediment is blocked by a concrete bike path bridge. (b) Grizzly Creek: Several in-channel fans formed several kilometers upstream of the Colorado River. No fan formed at the outlet of Grizzly Creek. (c) Deadhorse Creek: relatively minor depositional fan forms upstream of the Colorado River. (d) Tie Gulch: In-channel fan develops upstream of a knickpoint.**



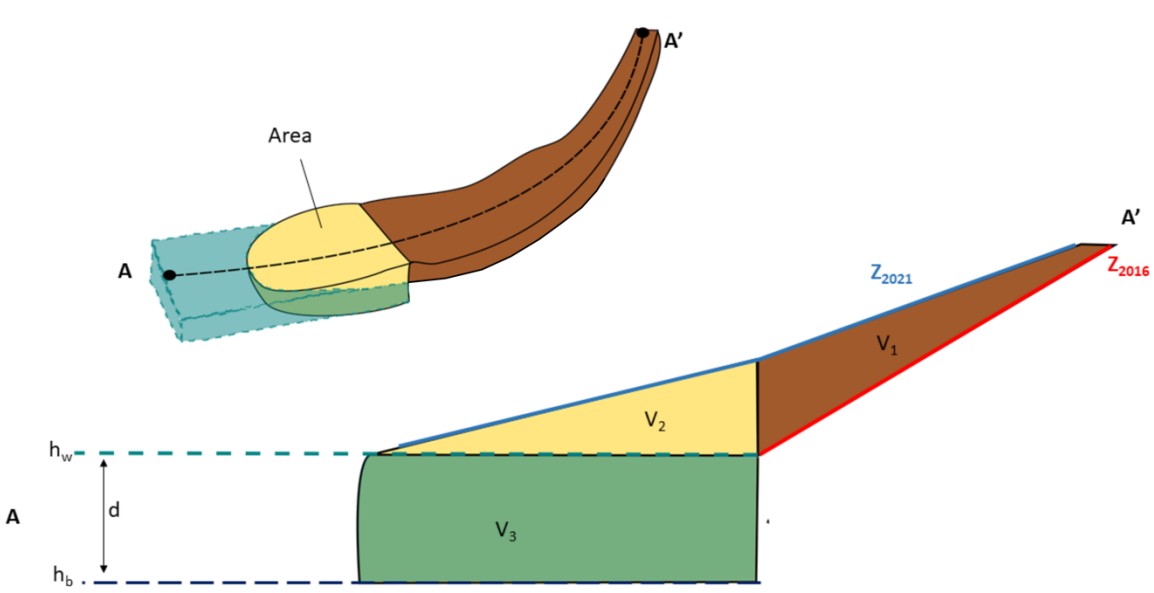

**Figure 6. (a) Oblique view and (b) Cross-sectional view of the approach used to estimate the volume of deposited sediment in the Colorado River due to unknown bathymetry. The volume of sediment deposition was divided into three zones. $V_1$ is the sub-areal zone, where a lidar difference can be used to estimate the volume. $V_2$ is the zone above the water surface. $V_3$ is the subaqueous zone, and the depth is estimated based on reports from CDOT maintenance crews.**



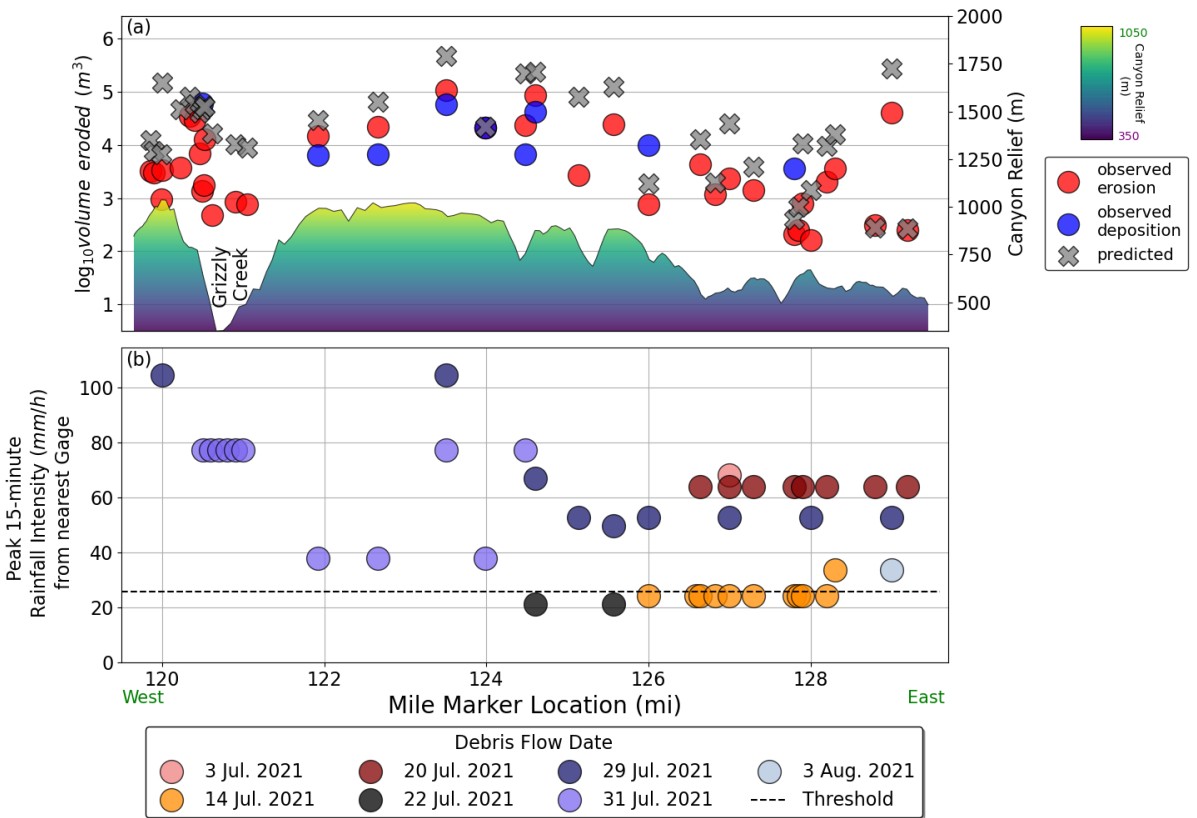

**Figure 7. (a) Observed sediment volume (erosional and depositional) and predicted sediment volume using *Equation 2* shown as a function of the mile marker location in Glenwood Canyon. For context, the canyon relief is shown on secondary axis. The relief profile indicates the elevation from the north side of the canyon approximately parallel to I-70. (b) Peak 15-minute rainfall intensity from the nearest representative gauge shown at locations identified by debris flow date. Note that channel polygons were associated with a mile marker location at the watershed outlet, as this was the preferred method of identification by CDOT personnel. In cases where the channel outlet did not directly intersect I-70, the outlet was simply translated onto a line segment representing I-70 to estimate a mile marker location.**





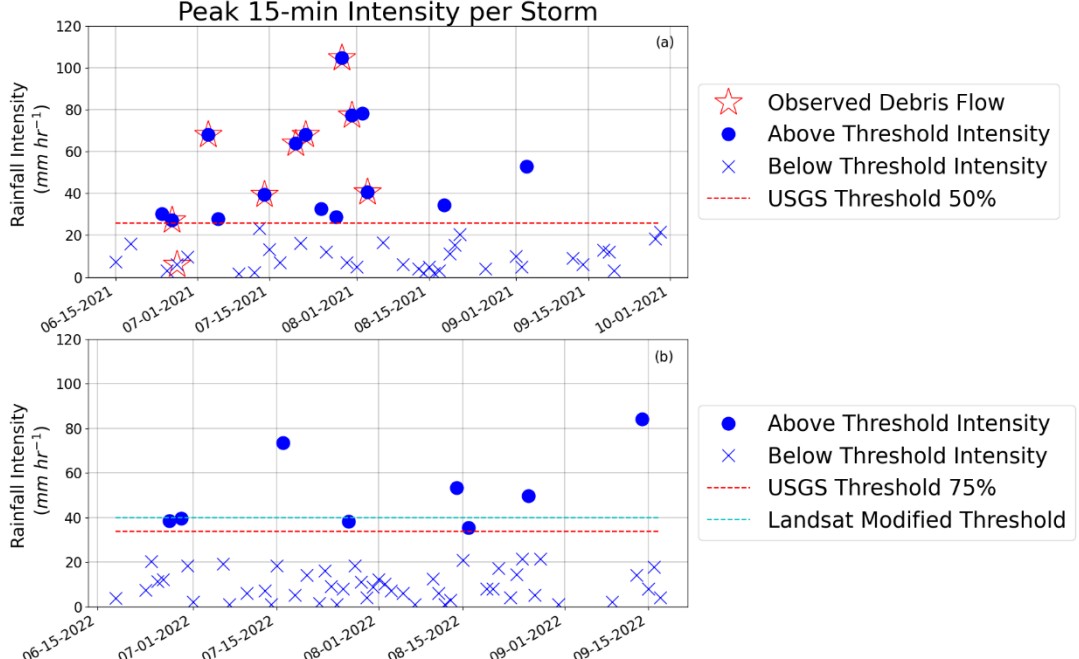

**Figure 8. (a) Maximum 15-minute rainfall intensity during each storm from the 11 rain gauges near the Grizzly Creek Fire perimeter during the 2021 monsoon. Rainfall rates below the modelled intensity threshold of 25.9 mm h$^{-1}$ (50% likelihood) are shown with a cross and those above the modelled threshold are shown with a circle. Any storms that produced a debris flow are indicated with a star. (b) Maximum 15-minute rainfall intensity during each storm from the 11 rain gauges near the Grizzly Creek Fire perimeter during the 2022 monsoon. The year 2 (75% likelihood) of 33.7 mm h$^{-1}$ was used as the USGS threshold. A recalibrated dNBR value from Landsat at the beginning of the monsoon season was used with the M1 model to develop an additional threshold of 40 mm h$^{-1}$ using the 50% likelihood of debris flow occurrence.**






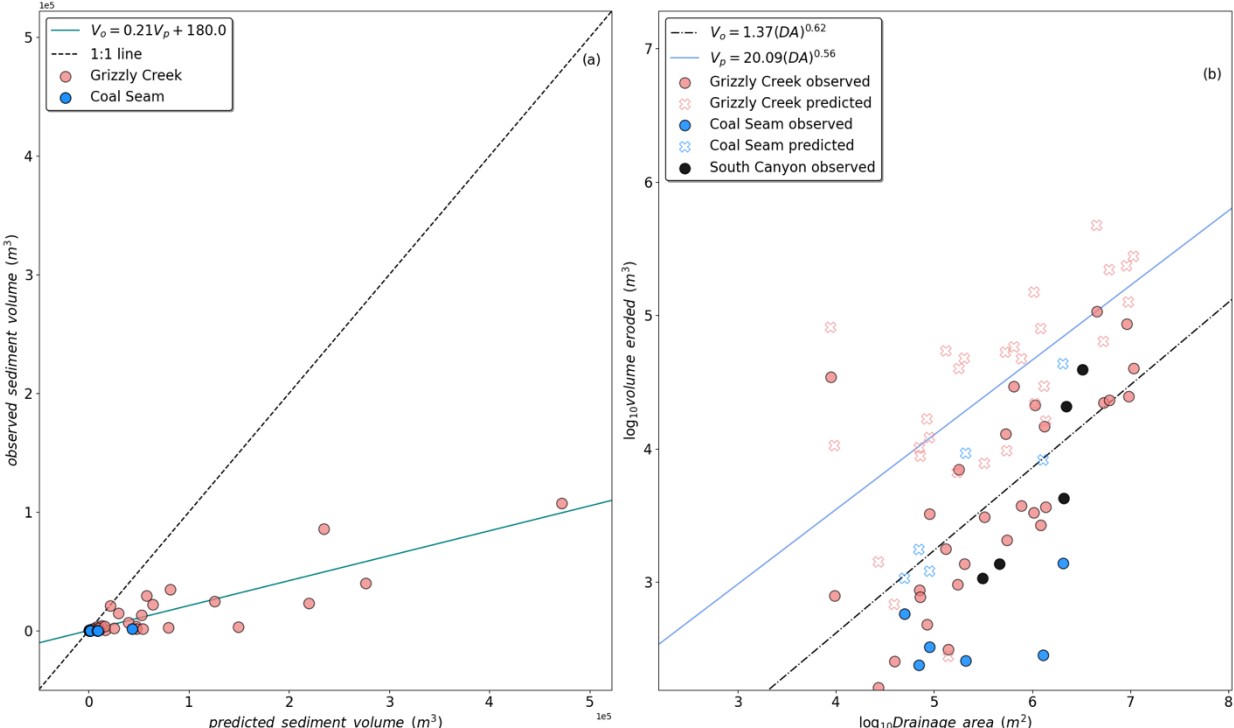

**Figure 9. (a) Observed sediment volume (erosional) versus predicted sediment volume using *Equation 2*. Linear trendline shows the relationship between the predicted ($V_p$) and observed ($V_o$) volume at the Grizzly Creek Fire. (b) A comparison of the total volume of observed sediment with predicted volume from *Equation 2* as a function of upstream drainage area (DA). The observation volumes represent the volume of erosion, upstream of a transition to deposition. Best-fit power law equations were fit to the observed and predicted data points for the Grizzly Creek Fire, respectively.**

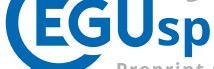

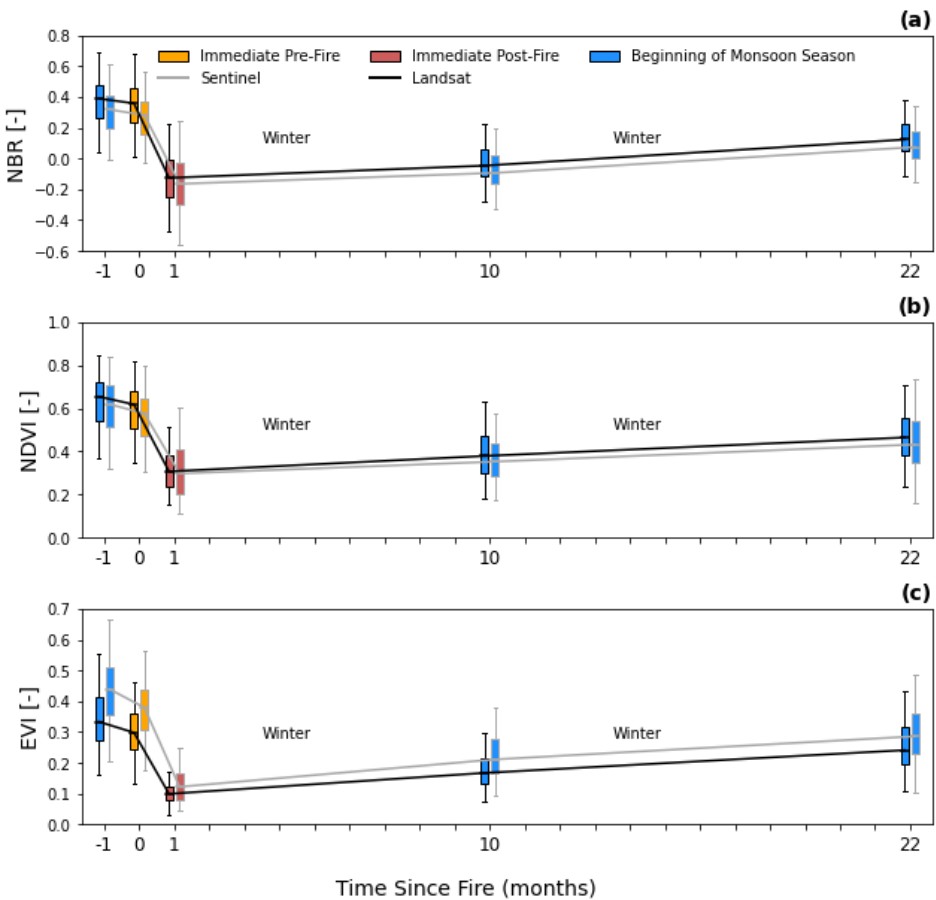

**Figure 10. Landsat- (black outline) versus Sentinel-derived (gray outline) measurements of the (a) Normalized Burn Ratio (NBR),**
**(b) Normalized Difference Vegetation Metric (NDVI), and (c) Enhanced Vegetation Index (EVI) for the 2020 Grizzly Creek Fire.**
**Boxplots summarize the distribution of the mean reflectance metric for each modelled basin (USGS 2022) that intersects the burn**
**area for the beginning of monsoon season (blue), immediately before the fire (orange), and immediately after the fire (red).**





**Figure 11. (a) View of debris flow paths and deposits in the Colorado River looking southwest on 18 August 2021. Arrows indicate new debris flow paths. (b) Sedimentation on the railroad in Glenwood Canyon on 1 August 2021. (c) Sedimentation on the lower deck of I-70 on 1 August 2021. (d) Photo of temporary damming of the Colorado River on 22 July 2021. Damming can be observed in the discharge record of the Colorado River on 22 July 2021. The upstream discharge in the Colorado River (Red) rises steadily in response to the closest measured 15-minute rainfall intensity (Blue). When the debris flow at Devil's Hole temporarily dammed the Colorado River, a drop in discharge is observed at the downstream river gauge (Cyan).**



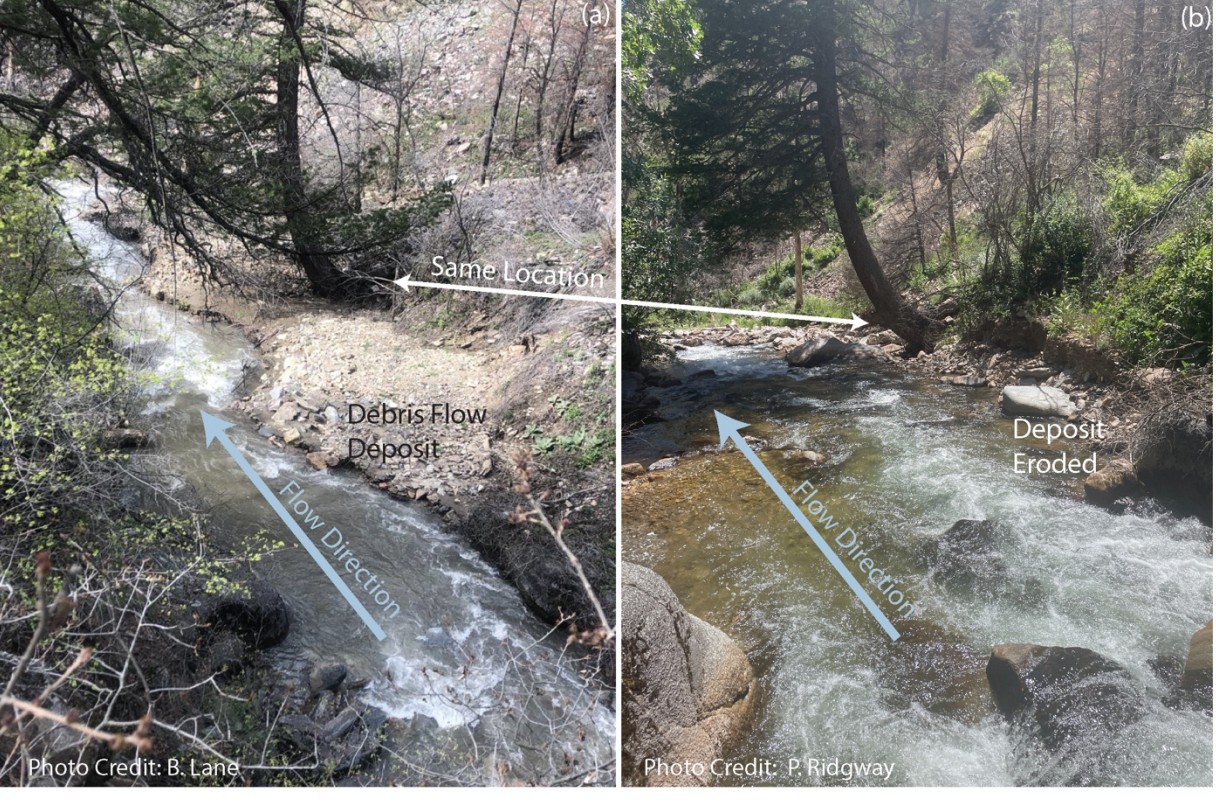

**Figure 12 (a) Small debris flow fan in Grizzly Creek at the beginning of the 2022 snowmelt season. (b)Vertical and horizontal incision of the same fan after the snowmelt runoff.**





**Tables**

**Table 1. Rain gauges deployed in and around the Grizzly Creek burn scar, operated by the USGS Colorado Water Science Center (USGS WSC), the Colorado Department of Transportation (CDOT), or the USGS Landslide Hazards Program (USGS LHP).**

| Rain Gauge Name | Owner | Station ID | Data Start | Data Stop | Data Gap |
|---|---|---|---|---|---|
| Cinnamon Creek Complex | USGS WSC | GCTC2 | 19 Jul. 2021 | present | 7/29/21 to 8/12/21 |
| Cinnamon Creek | USGS WSC | GCCC2 | 19 Jul. 2021 | present | No Gap |
| Deadmans Creek | USGS WSC | GCDC2 | 14 Jul. 2021 | present | 7/22/21-7/26/21 |
| No Name | USGS WSC | GCNC2 | 15 Jul. 2021 | present | 7/28/21 to 8/12/21 |
| Windy Point | USGS WSC | GCIC2 | 12 Jul. 2021 | present | No Gap |
| East Fork Dead Horse Creek | USGS WSC | GCEC2 | 13 Jul. 2021 | present | No Gap |
| Coffee Pot | USGS WSC | GCFC2 | 13 Jul. 2021 | present | No Gap |
| Bair Ranch | CDOT | N/a | 30 Jun. 2021 | present | No Gap |
| USGS_gc_1 | USGS LHP | N/a | 17 Sept. 2020 | present | No Gap |
| USGS_gc_2 | USGS LHP | N/a | 17 Sept. 2020 | present | No Gap |
| USGS_gc_3 | USGS LHP | N/a | 17 Sept. 2020 | present | No Gap |



**Table 2. Table of depositional fans with known storm triggering dates. For fans deposited within the Colorado River, CDOT**
**estimates of the sub-aerial sediment depth are provided.**

| Name of Fan | CDOT Depth Estimate Below Water Surface (m) | Total Fan Volume Estimate (m³) |
|---|---|---|
| Blue Gulch | 3.2 | 57,000±900 |
| Deadman Gulch | 1.5 | 6,800±400 |
| Devil's Hole | 4.6 | 42,000±700 |
| Maneater Gulch | 1.5 | 6,300±400 |
| Unnamed at Mile marker 124 | 2.4 | 21,000±700 |
| Wagon Gulch | 2.1 | 6,700±300 |
| Grizzly Creek (all fans) | n/a | 59,800±7500 |