# Peer review of "Evaluating Post-Wildfire Debris Flow Rainfall Thresholds and Volume Models at the 2020 Grizzly Creek Fire in Glenwood Canyon, Colorado, USA"

_EGUsphere, 2023_

## Author Comment (AC1)

**Note to editor/reviewers. Plain text represents the original comment, and bold text represents the response.**

Reviewer 1:

The paper presents a valuable exploration of the sources of error and potential improvements of common predictive methods for debris flow parameters. I have a few suggestions below:

Figure 3 - isn't this backwards for (b)? Blue is deposition and red is erosion?

[Figure]

**Figure 1. (a) Longitudinal Profile of the Blue Gulch watershed. (b) Longitudinal Profile of the Devil's Hole watershed In both (a) and (b) red dots and blue dots represent erosion/deposition, respectively. Areas without a dot did not experience change beyond the level of detection. (c) Measured deposition and erosion in the Blue Gulch watershed. (d) Measured deposition and erosion in the Devil's Hole watershed. In both (c) and (d) local slope is shown on the secondary y-axis.**

**One thing that may be making this confusing is that I didn't indicate the flow direction. I have now done this on the figure. The colors in Figure 3b are correct. In figure 3b flow goes from ~4800 m to 0m (this is the outlet), as flow moves from the upper reaches you see lots of erosion (red). There is a little bit of deposition (blue) at a low sloping spot around 3500 m, but the channel is mostly erosional (red) until the slope flattens out around 500 m. This can also be visualized in figure 3d. But in both cases the blue color is equal to deposition, and we are showing that as a positive value. We also added this text to make it more clear:**

**"Using the two lidar datasets (1 m pixel), we subtracted the elevation of the post event lidar from the pre event lidar ($Z_{2021} - Z_{2016}$) to create a DoD map of erosion and deposition throughout the Grizzly Creek burn area (Figure 2 and Figure 3)."**

**Based on that method readers should be able to see that wherever the post event data is positive, it should be equal to deposition, and those values are shown in blue in the legend.**

Figure 5 - I'm still confused. It makes sense that blue, representing positive lidar differences., should be positive.

**If you look in the legend of Figure 5, all shades of blue are positive indicating deposition, with the dash in-between just showing the range of values. All shades of red are negative indicating erosion.**

Figure 6 - why would V3 be rectangular?  Shouldn't it be trapezoidal?

**This is a very good question. We simply don't have any way to estimate the slope angle reasonably. No bathymetry exists. Additionally, we tried propagating the angle of the slope bank into the river, but that resulted in unrealistic depths. In the end, we thought that the most honest approach would be to use the field estimates of depth that we have. Those estimates are shown in Table 2.**

Figure 9 - Traditionally, shouldn't the observed (known) quantify be on the x-axis and the predicted (unknown) be on the y?

**This is a good point.  Typically, when fitting a model, you would definitely do it that way. In this case the predicted quantity of sediment volume is also known, so we are not looking at how an unknown varies as a function of the known. The reason we plotted the data in this way, is so that the coefficient of the line is what you multiply the predicted value by to get a corrected estimate. For example, taking the volume from the Gartner equation and multiplying by 0.21, as shown in our equation in the plot, is a little bit easier than if we plotted the opposite way because you would have to swap the sides of the equation to get a correction factor. It's not really hard, but we wanted to make the process as easy as possible.**

**I think part of this confusion is that we use the term "predicted", and create a variable called Vp. We have now changed the variable to Vg, representing Volume from the Gartner equation. In addition, we have changed to language throughout to say "estimated" volume rather than "predicted" volume. We think this will help clarify this issue for readers.**

Figure 11 - "Damming can be observed in the discharge record of the Colorado River on 22 July 2021. (e)"

**Good catch, thank you. Changed to have (e) at the beginning of the sentence now, which is now consistent with the rest of the caption.**

**"Figure 11. (a) View of debris-flow paths and deposits in the Colorado River looking southwest on 18 August 2021. Arrows indicate new debris-flow paths. (b) Sedimentation on the railroad in Glenwood Canyon on 1 August 2021. (c) Sedimentation on the lower deck of I-70 on 1 August 2021. (d) Photo of temporary damming of the Colorado River on 22 July 2021. (e) Damming can be observed in the discharge record of the Colorado River on 22 July 2021. The upstream discharge in the Colorado River (Red) rises steadily in response to the closest measured 15-minute rainfall intensity (Blue). When the debris flow at Devil's Hole temporarily dammed the Colorado River, a drop in discharge is observed at the downstream river gauge (Cyan)."**

Line 61 - consider changing to "more frequent and higher overland flow"

**Change to "increased likelihood of overland flow" because the frequency is dictated by the storms.**

Line 92 - "are applicable" versus what?  "No longer than the first two years"?  Might help to clarify

**Attempted to clarify this sentence by changing the text to say: "This work explored whether the current USGS operational models for debris-flow rainfall thresholds and volume successfully predicted debris flow occurrence and volume, respectively, at our study site during the first two years following wildfire."**

Line 116 - capitalize "Quaternary"?

**Changed.**

Section 3.3 - you might comment on the fact that you used data fro rainfall gauges close to the measured debris flow events, whereas Gartner, et al, and M1 model relied on a more scattered network. In essence, they were forced to rely on widely spread input data while you have the advantage of comparing to local data. I would like to hear your thoughts on this difference. Clearly, it is better to you more local data, but what can you say about the need in other cases to rely on general data as in Gartner, et al.?

**Regarding the distance from a rain gage, the Gartner et al., 2014 paper uses gages that are within 2km of a watershed, and the Staley et al., 2017 paper uses gages that are within 4 km of a watershed. In this study, our gages are on average 4.2 km from the watersheds of interest, but we have a maximum distance of 6.8 km. I have amended the text to indicate this now:**

**"The maximum distance between an observation and its associated gauge was 6.8 km, the minimum distance was 0.2 km, and the average distance was 4.2 km (Rengers et al., 2023b). These distances are similar to other studies. Staley et al. (2016) used rain gauges within 4 km of an observation to generate the debris-flow inventory used to develop the M1 model, and Gartner et al. (2014) used distances of 2 km to assign storms to debris-flow activity."**

This leads to a bigger issue. The Gartner et al. and M1 models rely on a dataset with limited accuracy, but smoothed over an area and temporally. You are using more accurate data, which one would expect to produce better results, but in some ways compares apples to oranges. For instance, earlier volume prediction models relied on StatsGo or other generalized soil data for things like Liquid Limit. More accurate liquid limit measurements would not improve that model because they represent micro-data that is very different from the area-averaged data that the model used. The area-averaged data is not very good, but simply adding focused accurate data does not improve the roughness of the model. I want to make sure that your analysis does not fall in the same trap - providing more precision on a dataset that lacks in accuracy. I think it would help to comment on this disconnect.

**The equation developed in Gartner et al., 2014 to predict postfire debris-flow volumes in the first year after fire was based on data that had considerable uncertainty. The methods that were used in that paper to obtain volume estimates were the following:**

**"Volumes of sediment removed from debris- retention basins were measured by counting the number of trucks filled during the cleanout of a debris-retention basin or by comparing field or aerial photographic surveys of full and empty debris-retention basins."**

**The final multiple linear regression model, referred to as the "Emergency Assessment Model" in the Gartner et al., 2014 paper has a residual standard error of 1.04 (the equation predicts ln(Volume) and**

thus this value has no unit) and the Gartner et al. 2014 training data generally fit within bounds of one order of magnitude of the predictions. I think our data are likely more accurate and precise than this because I imagine that the uncertainty associated with counting trucks could be quite high. However, we don't have a good measure of the uncertainty from the approach used in Gartner et al., 2014. I don't think this impacts the results of this manuscript though. Functionally, the "Emergency Assessment Model" was developed to estimate volumes, and users take the volumes from that equation to make predictions. Although the data we are comparing may be higher precision and accuracy, I'm not sure that has any implications on the model. We are not trying to re-calibrate the model, and that is where I think we would fall into the trap that you point out. I think we would need more observations of volume across the Colorado region to attempt a model recalibration.

One addition that I have now made, is that I added a band of plus or minus 2*standard error, to Figure 9 for reference (see below) because 95% of the data should fit within this. Also, this is close to +/- one order of magnitude and the Emergency Assessment model observations fit within plus or minus one order of magnitude.  So we now give readers a sense of the range that they should consider when looking at the volume estimates from Gartner et al., 2014.

[Figure]

Line 195ff - even more important than the side of the canyon (north or south) and the proximity, is the elevation of the rain gauge, to account for orographic effects.  Can you include a comment on this?

I agree that both elevation and the side of the canyon are important. Our mapping of debris-flow initiation showed that most of the debris flows started toward the canyon rim. In terms of elevation, this puts the initiation points closer to the elevation of the gauges, even if they are not as close distance-wise. Furthermore, due to the gage locations, we had to use the data available and didn't often have the luxury of choosing between both canyon side and gauge elevation. I made a map of the initiation points and the rain gauges that I have now added to the supplement.

[Figure]

**Figure S1. Map showing initiation locations of debris flows with respect to rain gauges and verified debris-flow deposits.**

Line 237 - was there notable deposition in the form of levees along the flow channel?

**We did see levees during some field investigations. Here is a photo, that I have now added to the supplement. I have also added the following text: "Some deposition was observed in the field in the form of levees, however, levees were not systematically mapped in the lidar because they were often smaller than the pixel cells and therefore difficult to verify without field verification."**

[Figure]

Flow
Direction

Debris-Flow
Channel Cross Section

Coarse-grained Levee

**Figure S4. Debris-flow levee with coarse grains shown adjacent to a debris-flow channel. The largest grain sizes in the levee were approximately 30 cm.**

Line 255 - not clear to me how V1 differs from V2. They both sound like subaerial deposition. Perhaps distinguish between them better? Also, I would like to see error or confidence estimates on your method of calculating the total volume.

**Yes we have now revised the text to make it more clear how we differentiate V1 and V2.**

**The original text said: "Next, we divided the volume deposited in the Colorado River into two pieces. V2 is the subaerial volume above the river water surface elevation (WSE) at the time of the post-event lidar collection."**

**The new text has been changed to better explain that V2 is simply the volume of sediment that sits in the area that used to be fully occupied by the river before the fan was deposited. The new text says:**

**"We divided the volume deposited in the Colorado River into two pieces. These volumes are stacked (one on top of the other) in an area that was fully occupied by the Colorado River prior to fan deposition. $V_2$ is the subaerial volume above the river water surface elevation (WSE) at the time of the post-event lidar collection."**

**As far as uncertainty, this is shown in Table 2 for all of the fans.**

Line 333 - I would argue that it is vegetation recovery and that sediment supply is not a limiting factor. Perhaps include a reference or two, and maybe shift the focus more on vegetation? Not to push my own papers, but you might check this one for an example of sediment supply independence:

Santi, P. and MacAulay, B., 2021, Water and Sediment Supply Requirements for Post-Wildfire Debris Flows in the Western United States, Environmental and Engineering Geology, vol. 27, pp. 73-85, doi.org/10.2113/EEG-D-20-00022.

**Good suggestion. Changed the text to say: "This observation likely results from vegetation recovery in susceptible basins as has been observed elsewhere (Santi and Macaulay, 2021)."**

Equation 6 - line 353 - wouldn't you normally want an equation that uses the observed value on the right side ((x-axis) to predict the values on the left side?

**Good point, we partially answered this in our response to Figure 9 above, but I want to just emphasize that this is a different situation and we have sought to make it more clear to readers by changing the language from "predicted" to "estimated" and we have changed the variable from Vp to Vg (see reasoning above). The advantage of our approach is that it allows us to use the slope (0.21) as the correction factor. That is, we multiply the volume from the Gartner equation (Vg) by 0.21 to obtain an estimate that better fits our observations. If we rearranged the equation, we'd have to divide the observed volume by the slope of the equation, and that is a little less straightforward.**

Lines 406-413 - It is great to see this vegetation-specific analysis. I've never seen this before.

**Thanks!**

Paul Santi

---

## Author Comment (AC2)

**Note to editor/reviewers. Plain text represents the original comment, and bold text represents the response.**

Reviewer 2:

Excellent, comprehensive analysis of debris flow activity after a fire. Findings reveal how well USGS operational models for debris flow rainfall thresholds and debris flow volumes perform for this fire location. Writing is clear, and figures are a nice combination of quantitative data and photos. My comments are mainly minor points of clarification.

1. Equation 1: what are the values of beta, C1, C2, and C3? Are they determined for this fire specifically or are there set values used across multiple fire locations?

**Good question. These do not differ based on fire locations, but they do change with rainfall duration. I have now tried to clarify this and have directed readers to the primary publication where Equation 1 was described. Here is the new text.**

**"Note the coefficients C1-C3 and β do not vary among fires or regions but differ based on rainfall duration. All values are shown in Staley et al. (2017)."**

2. Line 158, median I15T with overbar - the overbar on I15T seems to imply a mean value? If the overbar indicates a mean, what values are included in the mean? If it doesn't indicate a mean, what does the overbar represent? The median is the median threshold value computed for all basins?

**I have tried to clarify this, here is a new paragraph to clarify that we first generate thresholds for each basin then we summarize them to get a single fire-wide threshold for year 1 and year 2:**

**"Equation 1 is used to generate spatially explicit rainfall thresholds for individual channel segments or basins (< 8 km$^2$). However, in practice, managers can only use a single fire-wide rainfall threshold for warnings over a burn area. Therefore, to generate a single Year 1 threshold, we first estimated the 15-minute intensity rainfall threshold for all basins delineated by the hazard assessment (Figure 1) using** Error! Reference source not found. **and assuming p = 50% (P50). We then used the median value of all of the basins as the single fire-wide rainfall threshold for warning.  A similar method was used to estimate the Year 2 threshold, except we set p = 75% (P75) to estimate a Year 2 rainfall threshold, and then used the median rainfall threshold from all of the basins as the single fire-wide rainfall threshold. These probabilities were used to define a fire-wide 15-minute intensity rainfall threshold ($\overline{I15_T}$); however, the success rate of the P50 and P75 rainfall thresholds have not been rigorously tested. Therefore, in this study we compared the median P50 $\overline{I15_T}$ for all basins in the burn perimeter with the measured peak 15-minute intensity (I15) in 2021 ($\overline{I15_{T21}}$) and the P75 in 2022 ($\overline{I15_{T22}}$) to determine the performance of the P50 and P75 thresholds estimated using** Error! Reference source not found.**."**

3. The first sentence in section 3.2 seems to imply that the rain gauges are mapped in Figure 1 - but they are actually shown in Figure 2.

**Changed the text to clarify this: "These gauges (USGS_gc_1, USGS_gc_2, USGS_gc_3; Figure 2) were deployed specifically for the task of verifying the hazard assessment model (Figure 1)."**

4. What is the precision of the rain gauge measurements (what depth per tip?)

**I've now added this to Table 1.**

| Rain Gauge Name | Owner | Station ID | Data Start | Data Stop | Data Gap | Rain Gauge Model/Tipping Bucket Depth (mm) |
|---|---|---|---|---|---|---|
| Cinnamon Creek Complex | USGS WSC | GCTC2 | 19 Jul. 2021 | present | 7/29/21 to 8/12/21 | Vaisala WXT536/0.01 |
| Cinnamon Creek | USGS WSC | GCCC2 | 19 Jul. 2021 | present | No Gap | Vaisala WXT536/0.01 |
| Deadmans Creek | USGS WSC | GCDC2 | 14 Jul. 2021 | present | 7/22/21-7/26/21 | Vaisala WXT536/0.01 |
| No Name | USGS WSC | GCNC2 | 15 Jul. 2021 | present | 7/28/21 to 8/12/21 | Vaisala WXT536/0.01 |
| Windy Point | USGS WSC | GCIC2 | 12 Jul. 2021 | present | No Gap | Vaisala WXT536/0.01 |
| East Fork Dead Horse Creek | USGS WSC | GCEC2 | 13 Jul. 2021 | present | No Gap | Vaisala WXT536/0.01 |
| Coffee Pot | USGS WSC | GCFC2 | 13 Jul. 2021 | present | No Gap | Vaisala WXT536/0.01 |
| Bair Ranch | CDOT | N/a | 30 Jun. 2021 | present | No Gap | Vaisala RG13H/0.02 |
| USGS_gc_1 | USGS LHP | N/a | 17 Sept. 2020 | present | No Gap | HOBO RG3M/0.02 |
| USGS_gc_2 | USGS LHP | N/a | 17 Sept. 2020 | present | No Gap | HOBO RG3M/0.02 |
| USGS_gc_3 | USGS LHP | N/a | 17 Sept. 2020 | present | No Gap | HOBO RG3M/0.02 |

5. line 277, "If there were multiple storms that were triggered" - do you mean multiple storms that triggered debris flows?

**Yes, thanks for catching that. The new text says: "If there were multiple storms that triggered debris flows in the same watershed,…"**

6. Figure 1: I would have found it helpful to see the locations of debris flows on this figure rather than on a separate figure.

**Good Suggestion. I have now added the locations of the debris flow points to Figure 1. I didn't remove them from Figure 2, as I think they are useful reference points in both figures.  I also changed what I was showing in Figure 1 to something more relevant. Instead of showing the likelihood for debris flow initiation given a 15-minute rainfall intensity of 24 mm/hr, I am now showing the 15 minute rainfall intensity threshold assuming a 50% likelihood of debris flow initiation. The median value from all of these basins is used to estimate the Year 1 firewide threshold. I have also updated the caption to reflect this:**

[Figure]

**Figure 1. USGS debris-flow hazard assessment produced for the Grizzly Creek Fire perimeter using** Error! Reference source not found.**. The 15-minute rainfall intensity threshold is shown for each basin, assuming a likelihood of 50% (P50). The median value from all of these basins is used to estimate the Year 1 15-minute rainfall intensity threshold for the entire burn area.**

7. Figure 2b: I am not seeing deposition at the end of this debris flow track. For both b and c, consider adding an arrow to show flow direction.

**I've now added an arrow to show flow direction on Figure2b and 2c. As far as not seeing the deposition, this is a little hard it shows up more clearly in Figure 3. For the reviewer, I have illustrated this below. There is a small blue spot to the right of the red erosion. I've zoomed in and I added a black circle to indicate the deposition location for the reviewer. For the readers of the paper, I think they can see this most clearly in Figure 3a and 3c.**

[Figure]

**Figure 2. (a)** Map showing the burn perimeter, rain gauges, and locations of debris-flow observations. **(b)** DEM of difference map showing erosion (red) and deposition (blue) in the lower half of the Blue Gulch drainage. **(c)** DEM of Difference (DoD) showing erosion (red) and deposition (blue) in the lower half of the Devil's Hole drainage.

[Figure]

8. Figure 5d: this doesn't look like a fan - just in-channel deposition?

**Good point. I have changed the figure to say "in-channel deposits" and adjusted the text throughout.**

[Figure]

**Figure 3. In-channel sediment deposits in large drainage basins. (a) French Creek: a fan forms upstream of the Colorado River, primarily because sediment is blocked by a concrete bike path bridge. (b) Grizzly Creek: Several in-channel deposits formed several kilometers upstream of the Colorado River. No fan formed at the outlet of Grizzly Creek. (c) Deadhorse Creek: relatively minor depositional fan forms upstream of the Colorado River. (d) Tie Gulch: In-channel deposits develops upstream of a knickpoint.**

9. Figure 7: this is a nice figure but kind of a lot to take in. I am curious about how the erosion and deposition volumes compare at sites where both of those measurements were collected, but it's hard to evaluate that in this figure - could erosion-deposition comparison be pulled out as a separate figure or subplot? Then just show erosion in this plot? Do each of the debris flow volume points correspond with a rain intensity point in the bottom graph? Could be useful to have a volume vs. intensity plot, as an alternate way to visualize the data.

**This is a really useful comment. Based on this comment I have created a new figure that I have added to the supplement to show erosion versus deposition, and I've added in the uncertainty in both erosion and deposition. I will add this in as a new supplemental figure. Here's a note to the reviewer that I will put in the caption. Many channels contributed to an in-channel deposit in Grizzly Creek, but for the purposes of comparing erosion and deposition, all of the erosional channels contributing upstream of the deposit were summed to compare with the deposit volume.**

[Figure]

**Figure S6. Comparison of all the erosional volumes in locations where we have known depositional volumes. Error bars represent the uncertainty. Note that many channels contributed to an in-channel deposit in Grizzly Creek, but for the purposes of comparing erosion and deposition, all of the erosional channels contributing upstream of the deposit were summed to compare with the deposit volume.**

**As for updating figure 7. I don't think it hurts to leave in the deposition. I like the suggestion about plotting erosion versus rainfall intensity, but I think it is misleading so I am not including this figure. Here's the problem. There were multiple rainstorms between the two lidar datasets. We know the timing of the debris flows, as we have highlighted in several figures, and for comparison with the Gartner Volume equation we use the peak 15-minute intensity or sum of peak 15-minute intensities in the case of multiple debris flows because that is consistent with their methodology. However, because there were multiple storms and in some cases multiple debris flows, a plot of erosion versus maximum rainfall intensity may not account for the multiple pulses of the erosion that took place due to the cumulative rainfall and therefore it may be misleading.**

10. Figure 8. As I understand the model, the threshold intensity to produce a debris flow will vary by basin. Yet here the symbols for "above threshold intensity" seem to be based on the median threshold across basins? Why not show the symbols based on the threshold intensity computed for each basin individually? I am also confused by the caption text "from the 11 rain gauges" - are there intensity values given for each gauge individually or are the values averaged across all gauges?

**Hopefully this is now clarified by the text that I changed in response to comment 2 above.**

11. Figure 9. The value of (b) is not clear to me, other than as a means to add the south canyon data. Visually the power law lines do not seem to fit the data well.

**Good question regarding the value of (b). The main thing to observe is that the volume increases monotonically with drainage area. Therefore, we could apply a simple correction factor to lower the estimated volumes and that should work across drainage basins. I point this out in this text:**

**"Nevertheless, because of the linear nature of the offset in the volume estimate (Figure 9), it may be possible to apply a linear correction to the estimated volumes (e.g., multiply the estimated volume by 0.21) to obtain a regionally corrected volume estimate."**

**As far as the power-law fit, we now show an $R^2$ value and 2*standard error (which should capture 95% of the points around the trendline) so readers can interpret goodness-of-fit metrics. Also note that most debris-flow volume data are fairly noisy. Take a look at Figure 2 in Gartner et al., 2014 for reference.**

---

## Author Comment (AC3)

**Note to editor/reviewers. Plain text represents the original comment, and bold text represents the response.**

Reviewer 3
This study uses a unique debris flow response dataset from a region with a paucity of post-fire debris flow information to test the USGS PFDF susceptibility and volume models commonly used in the western US for emergency management. The authors find preliminary support for the use of regional correction factors for the volume model and reveal potential drivers for reduced susceptibility 2 years following fire. Overall, the manuscript is clear in its objectives, well-written, well-supported, and presents important findings. I recommend some minor revisions to improve clarity and make some suggestions for presentation of results that the authors could consider.

 **Thank you!**

Lines 116-117: Authors refer to Fig 1 when describing Quaternary-aged ("quaternary" should be capitalized, too) landslides but it is very difficult to see them in the hillshade as it is currently presented. Perhaps another figure in supplements if authors would like to show them, annotations directly on the plot highlighting the slides, or just no reference to Fig 1. Also, I don't know which watershed is Devil's Hole based on Fig 1.

**Yes good point. I have now capitalized Quaternary, and I also removed the reference to figure 1. The way I had written it, a reader might have expected to see landslides mapped in figure 1. We don't have room for that, and this is really just an auxiliary background information, not a key point of the study. So I've removed the reference to figure 1 so that readers don't expect to see a figure that shows mapped landslides. To see that figure, they can follow the reference that I have in the sentence. Finally, to see the location of Devil's Hole watershed, see the outline of Figure 2c in Figure 2a.**

Line 170: What are the models of tipping bucket rain gauges installed and their corresponding measurement resolutions?

**I've now added this to Table 1.**

**Table 1. Rain gauges deployed in and around the Grizzly Creek burn area, operated by the USGS Colorado Water Science Center (USGS WSC), the Colorado Department of Transportation (CDOT), or the USGS Landslide Hazards Program (USGS LHP).**

| Rain Gauge Name | Owner | Station ID | Data Start | Data Stop | Data Gap | Rain Gauge Model/Tipping Bucket Depth (mm) |
|---|---|---|---|---|---|---|
| Cinnamon Creek Complex | USGS WSC | GCTC2 | 19 Jul. 2021 | present | 7/29/21 to 8/12/21 | Vaisala WXT536/0.01 |
| Cinnamon Creek | USGS WSC | GCCC2 | 19 Jul. 2021 | present | No Gap | Vaisala WXT536/0.01 |
| Deadmans Creek | USGS WSC | GCDC2 | 14 Jul. 2021 | present | 7/22/21-7/26/21 | Vaisala WXT536/0.01 |
| No Name | USGS WSC | GCNC2 | 15 Jul. 2021 | present | 7/28/21 to 8/12/21 | Vaisala WXT536/0.01 |
| Windy Point | USGS WSC | GCIC2 | 12 Jul. 2021 | present | No Gap | Vaisala WXT536/0.01 |

| | | | | | | |
|---|---|---|---|---|---|---|
| East Fork Dead Horse Creek | USGS WSC | GCEC2 | 13 Jul. 2021 | present | No Gap | Vaisala WXT536/0.01 |
| Coffee Pot | USGS WSC | GCFC2 | 13 Jul. 2021 | present | No Gap | Vaisala WXT536/0.01 |
| Bair Ranch | CDOT | N/a | 30 Jun. 2021 | present | No Gap | Vaisala RG13H/0.02 |
| USGS_gc_1 | USGS LHP | N/a | 17 Sept. 2020 | present | No Gap | HOBO RG3M/0.02 |
| USGS_gc_2 | USGS LHP | N/a | 17 Sept. 2020 | present | No Gap | HOBO RG3M/0.02 |
| USGS_gc_3 | USGS LHP | N/a | 17 Sept. 2020 | present | No Gap | HOBO RG3M/0.02 |

Line 197: Shouldn't 4 km$^2$ be 4 km (distance not area)?

**Good catch. Changed to 4 km.**

Line 206: "see section 0" – there is no section 0?

**Thanks for pointing that out. That was an automated reference that failed. I have now changed it to section 3.4.2.**

Line 224: Were channel polygons hand-drawn or automatically extracted using a buffer around a flowline? Could be a nice detail to include.

**I clarified that they were mapped by hand at the end of this sentence: "For each debris-flow observation in our inventory, we mapped erosional and depositional areas in each channel with separate polygons by hand."**

Section 3.6: Recalculated dNBR post-recovery is a great idea and fits nicely with recent literature on quantifying vegetation recovery and its influence on debris flow susceptibility (such as Graber et al., 2023, link here: https://doi.org/10.1029/2023GL105101).

**Yes many of the authors of that study, were co-authors on this study and some of the ideas espoused in that paper, originated from the work done in this manuscript.**

Line 334: Was there evidence to support sediment exhaustion of the channels such as downcutting to bedrock? Could be good to include.

**We saw that in some places. Here's a photo, you can see the layered bedrock in the channel. This has been added to the supplement.**

[Figure]

**Figure S8. Photo from the upper portion of the canyon showing channelized debris-flow erosion down to bedrock indicating sediment depletion.**

Line 339: Not sure about using the term "nucleate" here and elsewhere (e.g. line 412) when referring to erosion/deposition in this context. I usually think of nucleation as a process that begins at one point and propagates outwardly, which I don't think describes what's happening here quite correctly. Could rephrase this as "initiate" or similar.

**Changed "nucleate" to "initiate" in both locations.**

Line 354-355: "changes between the debris flows and the lidar flight." Clarify. Do authors mean to say: "changes to the debris fans occurring between initial deposition and subsequent lidar flights" or something like this?

**Thanks, I changed the sentence to:**

**"The depositional volumes observed were less than the erosional volumes, as was expected due to sediment disturbance between the time of debris-flow deposition and the lidar flight (Figure 7)."**

Line 360: How were Coal Seam and South Canyon debris flow volumes estimated? Just curious if this could exert some uncertainty in a comparison of these earlier datasets to the lidar/fan based estimates for the Grizzly Creek PFDFs. It is promising that they roughly show similar area-volume scaling as the authors point out.

**Good question. The measurements for the Coal Seam and South Canyon debris flows were performed using the methods explained in Santi et al., 2008. Researchers made measurements in the field at cross-sections in channels measuring channel scour. I have added this sentence to the methods section:**

**"The volume data for the South Canyon and Coal Seam Fires were collected using the methods described by Santi et al. (2008) where researchers made measurements within channels estimating scour depth. The uncertainty differences between these field measurements and the lidar data are unclear; however, we estimate that the field measurements may be of a similar magnitude as the lidar (tens of centimeters)."**

Fig 8B: There were no debris flows produced in 2022, correct? Maybe add in this language to figure caption since it is a bit ambiguous as is (no red stars = none correct?).

**Added this sentence to the end of the caption:**

**"Note that no debris flows were observed in 2022."**

Fig 9A: The 1e5 scientific notation next to axis labels is too small. Consider blowing up this text or adding it alongside the units (e.g., 1e5 $m^3$).

**I increased the fontsize for clarity. See new version of figure below in response to next question.**

Fig 9B: I think the power law fits could be better coordinated with their respective point grouping colors – why have a Vp blue power law fit that does not match corresponding points with open red x's and Vo black dashed fit that does not match red circles. Also, these power law fits do not visually seem to fit their respective datasets very well. Additionally, it would be good to provide an estimate of goodness of fit metric ($R^2$) as well as p-values (or confidence intervals) for regression parameters (prefactor and exponent) to provide some degree of confidence of these fits.

**Some very good suggestions here. First, regarding colors, we agree, this can be improved. I think we have now simplified the colors to make it more clear. Here our revised logic for the new color choices. We are only fitting a line to the volumes predicted and observed at Grizzly Creek. We are then**

superimposing the Coal Seam and South Canyon data points to show how they compare. Therefore, because we use the red/pink color for Grizzly Creek data, we've made the power-law fit lines red/pink as well. We just use different line styles to differentiate them. We have also changed the color of the South Canyon Fire data to yellow from black, so that it isn't confused as having a relationship with the lines in Figure 9a.

As for powerlaw fit. We agree that the fit isn't great, and that the data don't conform to the assumptions of a normal distribution. Therefore, reporting a p-value doesn't really make sense. However, we do report the $R^2$ values on the plot now (see below) and show +/- 2 standard error, which should capture 95% of the data.

[Figure]

**Figure 1. (a) Observed sediment volume (erosional) versus estimated sediment volume using** Error! Reference source not found. **developed by Gartner et al. (2014). Linear trendline shows the relationship between the estimated ($V_g$) and observed ($V_o$) volume at the Grizzly Creek Fire. (b) A comparison of the total volume of observed sediment with estimated volume from** Error! Reference source not found. **as a function of upstream drainage area (DA). The observation volumes represent the volume of erosion, upstream of a transition to deposition. Best-fit power law equations were fit to the observed and estimated data points for the Grizzly Creek Fire, respectively.**

Additionally, for Fig 9B and the comparison between Coal Seam observed vs predicted (Line 360 earlier), do you see a similar ~4-fold overprediction from the Gartner et al. (2014) model? If it was close to this value, it further supports using this as a regional correction factor.

**Because of the incoming data for the South Canyon Fire, we don't have sufficient information to generate a volume estimate using the Gartner equation, which we state with this text:**

**"Limited rainfall data for the South Canyon Fire precluded the use of** Error! Reference source not found.**,…"**

Consequently, we can't say for certain the magnitude of the overprediction without applying Equation 2. However, we clearly see that the South Canyon observations are in-line with the observations made at the Grizzly Creek Fire. For the Coal Seam Fire, overpredictions range from 1.8x to 35x compared to the observations. Of the 6 observations, three of the overpredictions from the Gartner equation are <7.3x, and the other three observations are between 29x-35x. To highlight these differences to readers, we have added the following sentence to the discussion:

"The Coal Seam debris-flow volumes were overestimated between 1.8-35 times by Equation 2."